evolution, ecology

*Plasmodium*, transcriptional–translational feedback loop, periodicity, circadian rhythm, clock mutant

**Author for correspondence:**
Aidan J. O'Donnell
e-mail: aidan.odonnell@ed.ac.uk

# Host circadian clocks do not set the schedule for the within-host replication of malaria parasites

Aidan J. O'Donnell, Kimberley F. Prior and Sarah E. Reece

Institute of Evolutionary Biology, and Institute of Immunology and Infection Research, School of Biological Sciences, University of Edinburgh, Edinburgh, UK

 AJO, 0000-0003-3503-094X; SER, 0000-0001-6716-6732

Circadian clocks coordinate organisms' activities with daily cycles in their environment. Parasites are subject to daily rhythms in the within-host environment, resulting from clock-control of host activities, including immune responses. Parasites also exhibit rhythms in their activities: the timing of within-host replication by malaria parasites is coordinated to host feeding rhythms. Precisely which host feeding-related rhythm(s) parasites align with and how this is achieved are unknown. Understanding rhythmic replication in malaria parasites matters because it underpins disease symptoms and fuels transmission investment. We test if rhythmicity in parasite replication is coordinated with the host's feeding-related rhythms and/or rhythms driven by the host's canonical circadian clock. We find that parasite rhythms coordinate with the time of day that hosts feed in both wild-type and clock-mutant hosts, whereas parasite rhythms become dampened in clock-mutant hosts that eat continuously. Our results hold whether infections are initiated with synchronous or with desynchronized parasites. We conclude that malaria parasite replication is coordinated to rhythmic host processes that are independent of the core-clock proteins PERIOD 1 and 2; most likely, a periodic nutrient made available when the host digests food. Thus, novel interventions could disrupt parasite rhythms to reduce their fitness, without interference by host clock-controlled homeostasis.

## 1. Introduction

Biological rhythms are ubiquitous and allow organisms to maximize fitness by synchronizing behaviours, physiologies and cellular processes with periodicity in their environment. The value of coordinating with daily cycles in light/dark (LD) and temperature in the abiotic environment has long been appreciated, and the importance for parasites of coordinating with rhythms experienced inside hosts and vectors (i.e. the biotic environment) is gaining recognition [1–3]. For example, circadian rhythms in virulence enables the fungal pathogen *Botrytis cinerea* to cope with rhythmic immune defences in plant hosts [4,5], circadian control of macrophage migration provides incoming *Leishmania major* parasites with more host cells to invade at dusk than dawn [6] and host clocks control the ability of herpes and hepatitis viruses to invade cells and to replicate within them [7,8].

Malaria (*Plasmodium*) parasites exhibit periodicity in their development during cycles of asexual replication in red blood cells (the intra-erythrocytic development cycle; IDC). No known clock genes have been identified in *Plasmodium* genomes, but their gene expression patterns display some hallmarks of an endogenous clock [9,10]. Explaining how and why malaria parasites complete their IDC according to a particular schedule matters because cycles of asexual replication are responsible for the severity of malaria symptoms and fuel the production of transmission forms, and coordination with

the rhythms of both host and vector enhances parasite fitness [11–15]. Thus, insight into the IDC schedule may suggest novel interventions to disrupt parasite replication. Indeed, many antimalarial drugs have increased efficacy against specific IDC stages [16] and IDC stage-specific dormancy may facilitate parasite survival during antimalarial drug treatment [17].

The IDC lasts 24 h (or multiples of 24 h, depending on the species) and is characterized by progression through distinct developmental stages at particular times of day. For example, the timing of *Plasmodium chabaudi's* IDC transitions coincide with the time of day that murine hosts are provided with food [18,19]. Specifically, parasites remain in early IDC stages when hosts are fasting and complete the IDC at the end of the feeding phase. The foundation for explaining both why the IDC schedule benefits parasites and how it is controlled lies in discovering which of the myriad of host rhythms associated with the time of day that hosts feed also associate with the timing of the IDC schedule. Here, we use the rodent malaria parasite *P. chabaudi* to test whether the IDC schedule is coordinated with a host rhythm(s) that is driven by—or is independent of—the transcription–translation feedback loop (TTFL) that forms a major part of the host's circadian clock mechanism. The mammalian circadian clock operates via a core TTFL (which we hereafter call the core-TTFL clock) involving dimeric proteins that promote the expression of other clock proteins as well as the inhibition of themselves [20]. The feedback and degradation of these proteins forms an oscillator that is entrained via external daily stimuli (Zeitgeber, usually light) to keep the clock precisely tuned to environmental periodicity.

Core-TTFL clock-controlled processes undertaken by the host include many metabolic pathways relevant to IDC progression. For example, CLOCK and BMAL1 are involved in regulating blood glucose levels [21,22] and melatonin release, which are both implicated in IDC completion [18,19,23]. Alternatively, the IDC schedule could simply be aligned to the appearance of nutrients/metabolites made available in the blood as a direct consequence of food digestion (i.e. via processes not reliant on the host TTFL clock). Core-TTFL clock-controlled, and TTFL-independent products of digestion, could act in several non-mutually exclusive ways on the IDC, including: (i) impacting directly on IDC progression by providing essential resources for different IDC stages at different times of day, (ii) providing time-of-day information to the parasite to modulate its rate of development to maximize acquisition of such resources and (iii) act as a proxy for the timing (phase) of another important rhythmic factor that the parasite must coordinate with. Most of these scenarios, and most evidence to date [10–13,19,24], suggests the parasite possesses an ability to keep time.

To probe how core-TTFL clock-controlled host rhythms and host-feeding-related rhythms influence the IDC schedule, we apply time-restricted feeding (TRF) protocols to wild-type (WT) mice and clock-disrupted *Per1–Per2* double knockout mice (*Per1/2*-null) and compare the consequences for the IDC schedule of *P. chabaudi* infections initiated with either synchronous or desynchronized parasites. We hypothesize that if the IDC is scheduled according to a host-feeding-related rhythm alone, IDC completion will coincide with host feeding in WT and in *Per1/2*-null mice with a feeding rhythm (TRF), but that parasites become (or remain) desynchronized in *Per1/2*-null mice allowed to feed continuously.

By contrast, if feeding rhythms influence the IDC schedule via host TTFL-clock-controlled processes, parasites will only become (or remain) synchronous in WT mice, because they have both clocks and a feeding rhythm. We also test whether infection of TTFL-clock-disrupted mice has fitness consequences for both parasites and hosts. The mammalian TTFL-clock controls many aspects of rhythmicity in immunity [25], including the ability of leucocytes to migrate to the tissues [26] and the ability of macrophages to release cytokines [27]. Furthermore, rodents without functioning *Per2* lack IFN-γ mRNA cycling in the spleen (a key organ for malaria parasite clearance) and have decreased levels of pro-inflammatory cytokines in blood serum [28]. Thus, we predict that parasites will achieve higher densities, and hosts experience more severe disease, in *Per1/2*-null compared to WT mice.

## 2. Methodology

To test if rhythmicity in parasite replication is coordinated with the host's feeding-related rhythms and/or rhythms driven by the host's canonical circadian clock, we performed two experiments. First, we initiated infections with desynchronized parasites to test whether a host feeding rhythm alone is sufficient to restore synchrony and timing in the IDC (figure 1a). Second, we tested whether the loss of rhythmic host feeding leads to desynchronization of the IDC in infections initiated with synchronized parasites (figure 1b).

### (a) Parasites and hosts

Hosts were either WT C57BL/6 J strain or *Per1/2*-null clock-disrupted mice previously backcrossed onto a C57BL/6 J background for over 10 generations. *Per1/2*-null mice (kindly donated by Michael Hastings, MRC Laboratory of Molecular Biology, Cambridge, UK) derived from JAX strains #010831 (129S-Per1$^{tm1Drw}$/J) and #010492 (129S-Per1$^{tm1Drw}$/J) generated by David Weaver (UMass Medical School, MA, USA) have an impaired core-TTFL clock and exhibit no known circadian rhythms in physiology and behaviour. For example, their locomotor activity is arrhythmic when placed in constant conditions, such as constant darkness [29,30]. We housed all experimental WT and *Per1/2*-null mice (8–10 weeks old) at 21°C in DD (continuous darkness) 'free-running' conditions with constant dim red LED light for three weeks prior to, and throughout the duration of infections. This allowed sufficient time for the erosion of residual (ultradian or unconsolidated) rhythms that can persist when clock-disrupted mice enter DD conditions [29]. Note, we housed donor mice in LD cycle conditions to generate synchronous parasites for the initiation of experimental infections. As the period (the time taken for a rhythm to complete one full cycle) of our WT mice is very close to 24 h (23.8–23.9 h; electronic supplementary material) when placed in DD, these mice exhibit rhythms very similar to the LD conditions they were raised in. Therefore, we define subjective day (rest phase) for WT mice as 07.00–19.00 GMT and subjective night (active phase) as 19.00–07.00 GMT.

We fed all mice on a standard RM3 pelleted diet (801700, SDS, UK) with unrestricted access to drinking water supplemented with 0.05% *para*-aminobenzoic acid (to supplement parasite growth, as is routine for this model system). All mice were acclimatized to their feeding treatments (see below) for three weeks before and throughout infections and housed individually to avoid any influence of conspecific cage-mates on their rhythms. On Day 0, we infected each mouse with $5 \times 10^6$ *P. chabaudi* (clone DK) parasitized red blood cells administered

Proc. R. Soc. B 287: 20200347

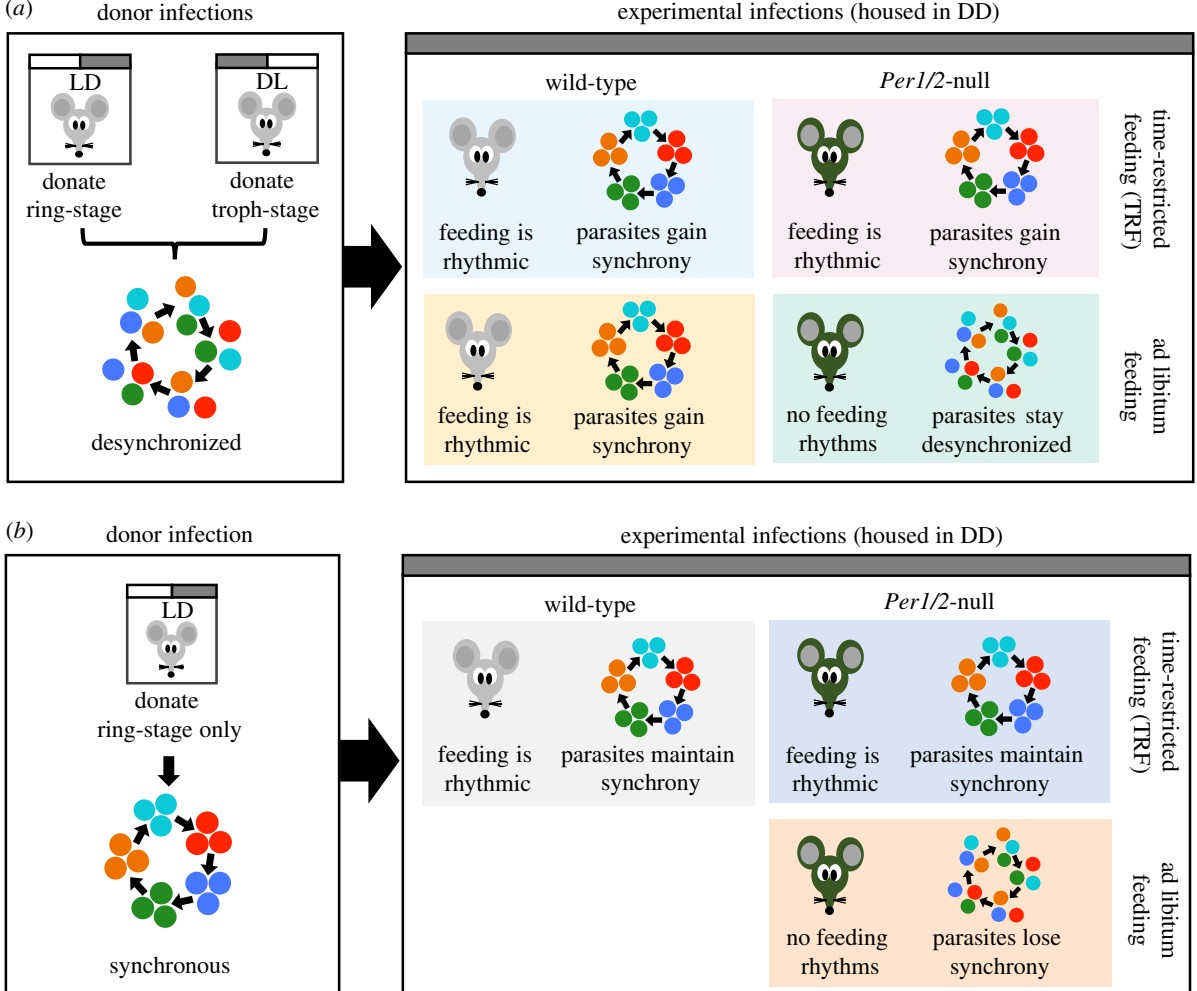

**Figure 1.** Experimental designs and predictions. Donor infections from mice housed in LD and/or DL were used to generate a desynchronized (*a*; ring stage + trophozoite stage parasites) and synchronous (*b*; ring stage parasites only) inocula for initiating experimental infections. WT or *Per1/2*-null clock-disrupted mice were given constant access to food (ad libitum) or fed on a TRF schedule in which food access was restricted to only 10 h d$^{-1}$. These mice were used as hosts for experimental infections and sampled every 4 h for 32 h on Day 5 and 6 PI. We predicted that desynchronized infections will become synchronous in mice in which feeding is rhythmic (both WT groups and *Per1/2*-null TRF) but will remain desynchronized in the ad libitum fed *Per1/2*-null mice due to a lack of host feeding rhythms. Furthermore, we expected the timing of parasites that become synchronous will match host feeding rhythms. Thus, parasites in WT ad libitum fed mice will follow the opposite schedule to parasites in WT TRF and *Per1/2*-null TRF mice (*a*). For infections initiated with synchronous parasites, we predicted that parasites will maintain synchrony in WT and *Per1/2*-null TRF groups and that the IDC schedule changes to match the timing of host feeding, but that synchrony will decay in parasites in ad libitum fed *Per1/2*-null mice, which lack feeding rhythms (*b*). (Online version in colour.)

via intravenous injection. All procedures complied with the UK Animals (Scientific Procedures) Act 1986 (PPL 70/8546).

## (b) Experimental designs

WT mice and *Per1/2*-null mice experienced a TRF schedule (fed for 10 h d$^{-1}$) or had continuous access to food ad libitum. The TRF mice remained in their cages during food provision and removal, to minimize disturbance, and food was provided/removed by changing the lid (which held the food) and sweeping the cage for stray pellets at the times of removal.

Despite continuous access to food, WT mice followed their normal free-running rhythms and fed primarily in their subjective night (19.00–07.00 GMT). Whereas, WT TRF mice fed only in their subjective day (09.00–19.00 GMT), causing temporal misalignment between rhythms controlled by the suprachiasmatic nucleus (SCN) and peripheral rhythms [19]. *Per1/2*-null TRF mice fed during the day (09.00–19.00 GMT) only experienced rhythms resulting from a set daily period of feeding (electronic supplementary material), whereas ad libitum *Per1/2*-null mice were arrhythmic (electronic supplementary material) due to continuous feeding. TRF feeding regimes do not cause caloric restriction because mice are given unrestricted access to food during their daily feeding window.

## (c) Experiment 1: can desynchronized parasites restore the intra-erythrocytic development cycle schedule in hosts with a feeding rhythm?

We generated four treatment groups of $n = 5$ mice (figure 1*a*): (i) WT ad libitum fed mice that naturally feed during subjective night; (ii) WT TRF mice fed only during subjective day; (iii) *Per1/2*-null mice fed ad libitum; and (iv) *Per1/2*-null TRF mice fed during the day. We initiated infections in all mice with a population of desynchronized parasites at 08.30 GMT by using an inoculum of a 50 : 50 mix of parasites 12 h apart in their IDC. Specifically, we mixed ring stages (donated from donors in a 12 : 12 LD cycle) and late trophozoite stages (donated from dark : light (DL) donors) (figure 1*a*).

## (d) Experiment 2: do parasites lose intra-erythrocytic development cycle synchrony in the absence of a host feeding rhythm?

We generated three groups of $n = 5$ mice (figure 1*b*): (i) WT TRF mice fed during subjective day; (ii) *Per1/2*-null TRF mice

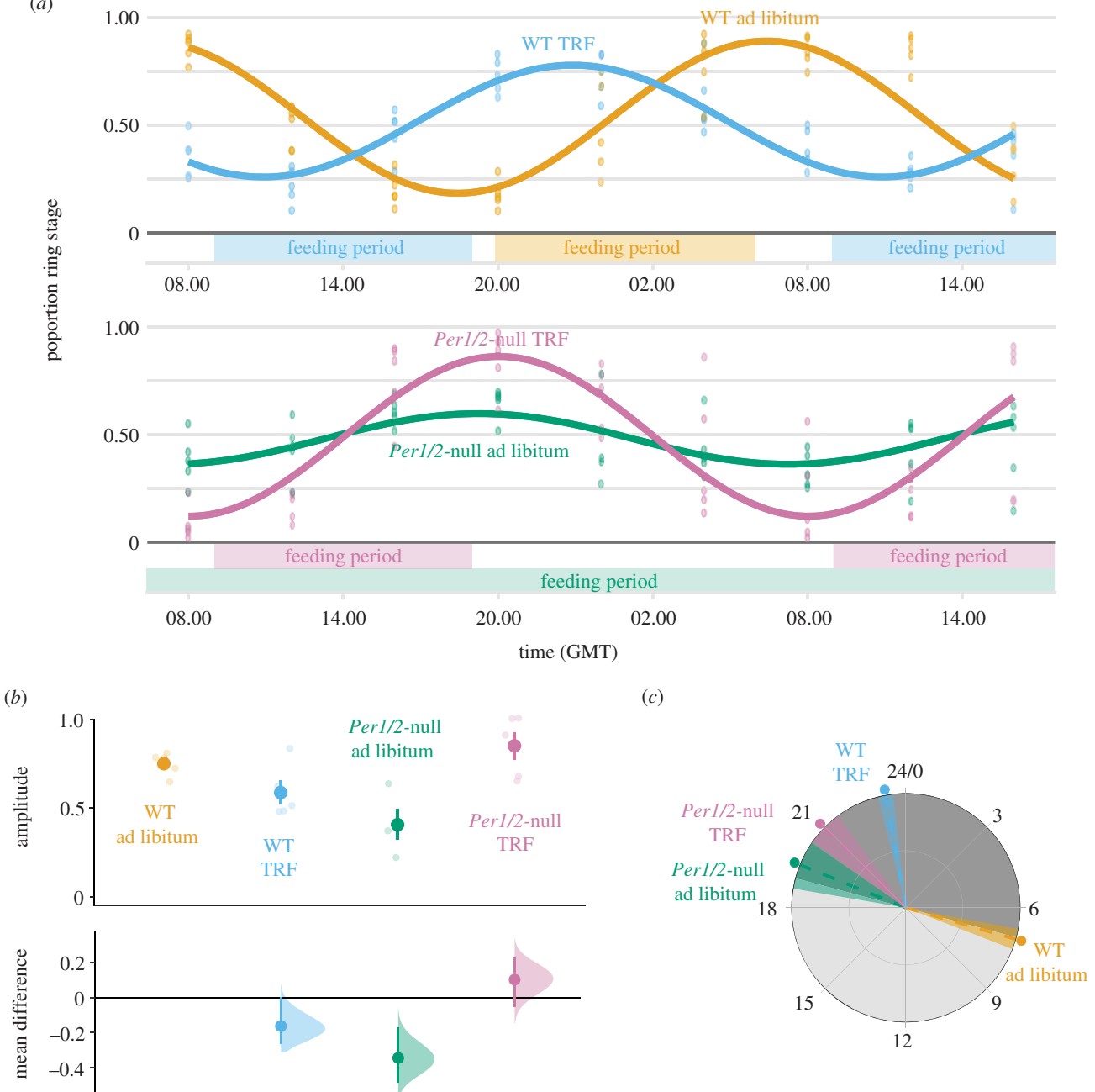

**Figure 2.** The IDC of desynchronized parasites becomes coordinated to host feeding-associated rhythms. Population cosinor model fits and data points from each individual infection (Day 5–6 PI) (*a*). Amplitude (*b*) and phase in hours (GMT) (*c*) were calculated from cosinor model fits from each individual mouse (lighter points) and then summarized as a mean ± s.e.m., points with error bars in (*b*), and circular mean ± s.d. point with dashed line and shading in (*c*). For amplitude (*b*), effect sizes relative to the 'WT ad libitum' group are plotted on the lower axes as a bootstrap sampling distribution (mean difference ± 95% CI depicted as a point with error bars). For all parts, infections in WT hosts are coloured orange and blue, and infections in *Per1/2*-null mice are coloured green and purple (*n* = 5 for the WT and TRF groups, *n* = 4 for *Per1/2*-null ad libitum group). TRF indicates 'time-restricted feeding' with food only available for 10 h each day (feeding period indicated above *x* axis in (*a*)). Grey shading in (*c*) represents active (dark shading; 19.00–07.00) and rest (light shading; 07.00–19.00) periods relative to WT mice in DD. (Online version in colour.)

fed during the day; and (iii) *Per1/2*-null with continuous (ad libitum) access to food. We infected all mice with a population of synchronous ring stage parasites early in the feeding period (which is 12 h out of phase to when rings stages peak in control infections (figure 1*b*)). Generating a mismatch between incoming parasites and the recipient host's feeding rhythm tests whether the IDC becomes rescheduled to match the feeding rhythm in the TRF groups. This avoids an outcome of the IDC being constrained and unable to change schedule obscuring the importance of a host-feeding rhythm, following the design in Prior *et al.* [19].

## (e) Sampling and data collection

We sampled all experimental mice at 4-hourly intervals for 32 h beginning at 08.00 (GMT) Day 5 to 16.00 Day 6 post-infection (PI). Previous work [19] revealed that synchronous parasites in infections initially mismatched to the host's feeding rhythm by 12 h (as we do in Experiment 2) exhibit a rescheduled IDC within 4 days. This phenomenon is verified here: the IDC became rescheduled in the WT TRF mice fed during subjective day (figures 2*a* and 3*a*). At each sampling point, we collected blood from the tail vein and quantified each IDC stage from

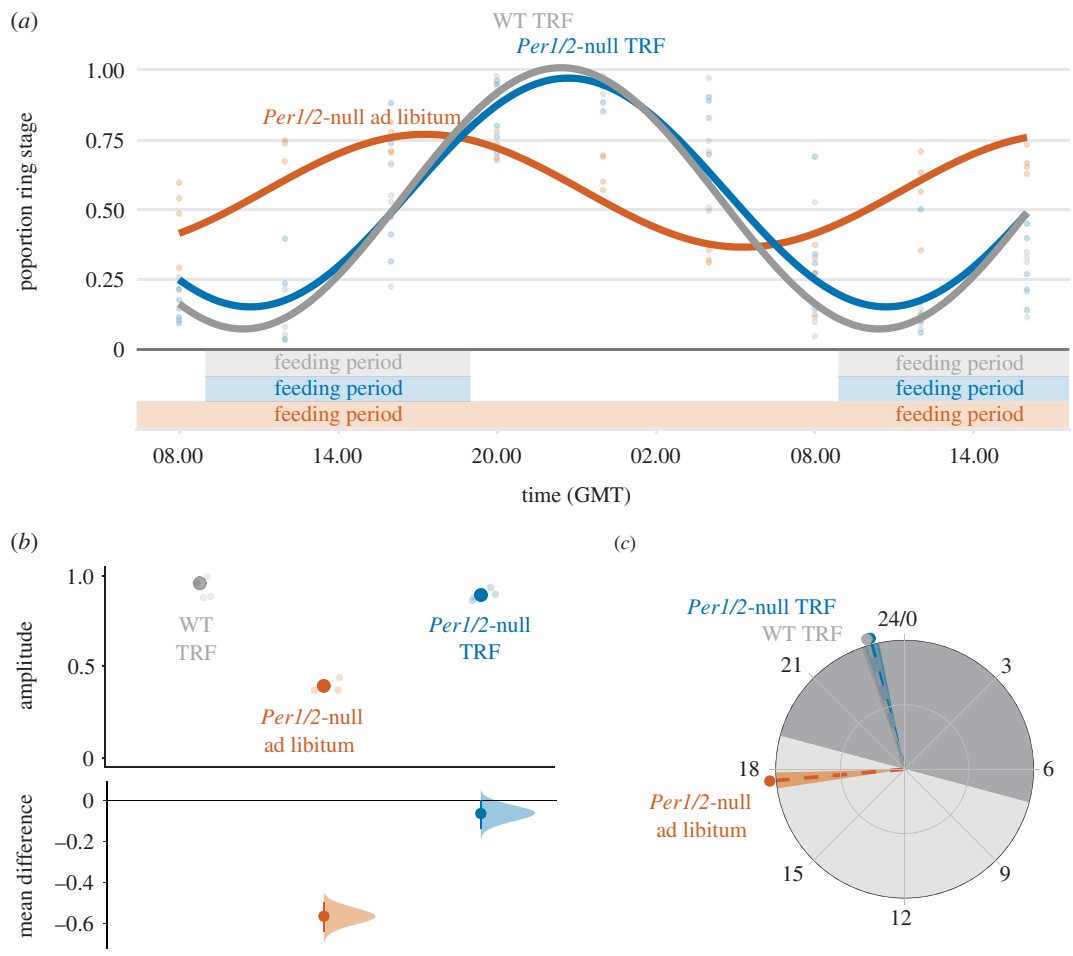

**Figure 3.** IDC synchrony is reduced in hosts without feeding-associated rhythms. Population cosinor model fits and data points from each individual infection (Day 5–6 PI) (*a*). Amplitude (*b*) and phase in hours (GMT) (*c*) were calculated from cosinor model fits from each individual mouse (lighter points) and then summarized as a mean ± s.e.m., points with error bars in (*b*) and circular mean ± s.d. point with dashed line and shading in (*c*). For amplitude (*b*), effect sizes relative to the 'WT TRF' group are plotted on the lower axes as a bootstrap sampling distribution (mean difference ± 95% CI depicted as a point with error bars). For all parts, infections in WT are coloured grey, and infections in *Per1/2*-null mice are coloured orange and blue (*n* = 5 for WT, *n* = 4 for TRF and *n* = 3 for *Per1/2*-null ad libitum group). TRF indicates 'time-restricted feeding' with food only available for 10 h each day (feeding period indicated above *x* axis in (*a*)). Grey shading in (*c*) represents active (dark shading; 19.00–07.00) and rest (light shading; 07.00–19.00) periods relative to WT mice in DD. (Online version in colour.)

thin blood smears, as is standard for measuring the IDC schedule [10–13,19,24]. Specifically, we characterized stages by morphology, based on parasite size, the size and number of nuclei and the appearance of haemozoin (as per [19,31]). We measured red blood cell (RBC) densities at each sampling time by flow cytometry (Z2 Coulter Counter, Beckman Coulter), and mouse weights on Day 2 PI and Day 6 PI at 16.00 GMT. All procedures were carried out in dim red LED light. Before infection, we characterized rhythms in locomotor activity (movement around the cage) and internal body temperature for all host genotype and feeding regime combinations (electronic supplementary material), and tested whether locomotor activity is a proxy for feeding events (electronic supplementary material). Data were analysed using Clocklab, CircWave, JTK_CYCLE and R (see electronic supplementary material, for details).

## 3. Results

### (a) Assumptions of the experimental designs

We first verified that WT mice exhibit rhythms in locomotor activity and body temperature, and also confirmed arrhythmic activity of *Per1/2*-null ad libitum fed mice (electronic supplementary material, Methods, and figures S1–S3). We then verified that locomotor activity can be used as a proxy

for feeding rhythms in *Per1/2*-null TRF mice (electronic supplementary material, Methods and figure S4).

### (b) Experiment 1: can desynchronized parasites restore the intra-erythrocytic development cycle schedule in hosts with a feeding rhythm?

We compared IDC rhythms in terms of synchronicity (amplitude) and timing (phase) of the proportion of parasites at ring stage (a morphologically distinct 'marker' stage after which all other parasite stages follow in a predictable manner) [19]. We do not estimate period due to the short sampling window. By Day 5–6 PI, the IDC of parasites in all WT mice and *Per1/2*-null TRF mice had become synchronized and scheduled to coincide with host feeding rhythms (figure 2*a*). Amplitude differed significantly between groups (figure 2*b*; genotype : feeding_regime: $F_{1,15} = 20.54$, $p < 0.001$). Specifically, parasites in *Per1/2*-null TRF mice had the highest amplitudes (mean ± s.e.m.: 0.85 ± 0.08) followed by WT ad libitum infections (0.75 ± 0.03), and then WT TRF infections (0.59 ± 0.07), with *Per1/2*-null ad libitum infections (0.41 ± 0.09) exhibiting approximately half the amplitude of parasites in hosts with feeding rhythms. Concomitantly, the timing of

peak ring stage proportion was explained by a host genotype : feeding regime interaction (figure 2c; electronic supplementary material, table S3): peaking in WT mice at the end of the host's feeding window (circular mean ± s.d. (hours GMT): WT ad libitum = 7.06 ± 0.35), and within 1–2 h of the end of the host's feeding window in TRF mice (WT TRF = 23.32 ± 0.27, Per1/2-null TRF = 20.96 ± 0.61). Despite the severely dampened rhythm, ring stages in ad libitum fed Per1/2-null mice peaked at 19.47 GMT (±0.83). See electronic supplementary material, for CircWave model fits, results from JTK_CYCLE, and mean effect sizes.

We also assessed whether anaemia and parasite performance varied between WT and Per1/2-null mice. Neither host genotype, feeding regime or their interaction significantly affected RBC loss (genotype : feeding_regime: $F_{1,16} = 0.27$, $p = 0.61$; host genotype: $F_{1,17} = 1.95$, $p = 0.18$; feeding regime: $F_{1,18} = 0.16$, $p = 0.70$), with hosts losing an average of $2.50 \pm 0.18$ s.e.m. $\times 10^9$ ml$^{-1}$ RBCs during the sampling period (electronic supplementary material, figure S5a and table S4). By contrast, host genotype had a significant effect on maximum parasite density ($F_{1,18} = 12.86$, $p = 0.002$) in which parasites infecting WT hosts achieved maximum densities approximately 40% higher than parasites infecting Per1/2-null mice (mean ± s.e.m. $\times 10^9$ ml$^{-1}$: WT = 1.69 ± 0.08, Per1/2-null = 1.22 ± 0.10; electronic supplementary material, figure S5b and table S4). Neither feeding regime ($F_{(1,17)} = 0.24$, $p = 0.63$) nor its interaction with host genotype ($F_{1,16} = 0.36$, $p = 0.56$) had an effect on maximum parasite density.

## (c) Experiment 2: do parasites lose intra-erythrocytic development cycle synchrony in the absence of a host feeding rhythm?

By Day 5–6 PI, the IDC of parasites in TRF mice had rescheduled to coincide with host feeding rhythms (figure 3a) verifying that sufficient time had been allowed for the IDC schedule to respond to the different perturbations of host TTFL-clock and feeding rhythms across treatment groups. IDC synchrony differed significantly between treatment groups (figure 3b; $F_{2,9} = 91.40$, $p < 0.001$), remaining high in TRF mice and dampening in Per1/2-null ad libitum mice. Specifically, ring stage amplitudes in TRF mice (mean ± s.e.m.: WT TRF = 0.96 ± 0.04, Per1/2-null TRF = 0.90 ± 0.02) were more than 50% higher than amplitudes for ring stages in Per1/2-null ad libitum mice (0.39 ± 0.02). In addition, the timing of the IDC varied across treatment groups (figure 3c; electronic supplementary material, table S3). Ring stages in TRF mice peaked 4 h after their host's feeding window (circular mean ± s.d. (hours GMT); WT TRF = 22.92 ± 0.22, Per1/2-null TRF = 23.01 ± 0.22), whereas ring stages with dampened rhythms in Per1/2-null ad libitum mice peaked 8 h earlier than in TRF groups (17.66 ± 0.24). See electronic supplementary material, for CircWave model fits, results from JTK_CYCLE, and mean effect sizes.

In contrast with infections initiated with desynchronized parasites, RBC loss varied across treatment groups of infections initiated with synchronous parasites ($F_{2,11} = 23.62$, $p < 0.001$; electronic supplementary material, figure S6a and table S4). WT TRF mice lost the most RBCs (30–57% more than both groups of Per1/2-null mice), and ad libitum fed Per1/2-null mice lost 20% more RBCs than their Per1/2-null TRF counterparts (mean RBC loss ± s.e.m. $\times 10^9$ ml$^{-1}$: WT

TRF = 3.57 ± 0.13, Per1/2-null TRF = 2.28 ± 0.15, Per1/2-null ad libitum = 2.73 ± 0.13). As we found for infections initiated with desynchronized parasites, maximum parasite density of infections initiated with synchronous parasites also varied across treatment groups ($F_{2,11} = 4.40$, $p = 0.04$). Parasites in WT TRF hosts achieved maximum parasite densities 24% higher than parasites in both groups of Per1/2-null mice (mean max. density ± s.e.m. $\times 10^9$ ml$^{-1}$: WT TRF = 1.89 ± 0.11, Per1/2-null = 1.51 ± 0.07; electronic supplementary material, figure S6b and table S4).

## 4. Discussion

Our results demonstrate that timing and synchrony of the malaria parasite P. chabaudi's IDC is not dependent on rhythms driven by the core-TTFL clock of hosts, and that parasites establish an IDC schedule in hosts with only rhythms associated with feeding. Our first experiment revealed that parasites within infections initiated with desynchronized parasites became synchronized in hosts with a feeding rhythm, and these infections exhibited a similar timing (reaching an average peak ring stage proportion of 86% within an hour after the feeding window ends). Furthermore, in ad libitum fed clock-disrupted hosts, which feed in many small irregular bouts across each 24 h period, the IDC remained desynchronized (ring stage proportion remaining at around 50% across all sampling points). Consistent with these phenomena, our second experiment revealed that the IDC of parasites in infections initiated with synchronous parasites remained synchronous and became coordinated to the timing of host feeding but only in hosts with feeding rhythms. Whereas in ad libitum fed clock-disrupted mice, the IDC rhythm became dampened (peak in ring stages dropping from approx. 100% to approx. 75%). Put another way, both experiments show that an IDC schedule emerges in hosts with a feeding rhythm independently of the host's core-TTFL clock, and the IDC rhythm is dampened in hosts without a feeding rhythm. We expect our findings to generalize across strains, given the similarities in the IDC schedules observed in this study and in Hirako et al. [18] and Prior et al. [31] which used different strains of P. chabaudi.

While the IDC rhythm of synchronous parasites inoculated into ad libitum fed clock-disrupted mice became dampened, it did not become fully desynchronized. There are two non-mutually exclusive reasons for this. First, there are likely to be developmental constraints acting on the duration of each IDC stage and the overall IDC length, independent of the influence of any host scheduling forces or parasite time-keeping abilities. If the minimum and maximum duration of the IDC is close to 24 h, or stage durations are similarly constrained, natural variation in IDC duration will take more cycles to fully erode synchrony than allowed in our experiment [32]. Determining how many cycles it will take a population of parasites to fully desynchronize is complex because the rate will be obscured by changes in density [33]. Thus, without sophisticated modelling that accounts for infection dynamics, it is difficult to determine whether desynchronization of the IDC, when hosts are in constant conditions, is due to a free-running oscillator belonging to the parasite [9]. Second, even in completely asynchronous infections, the expansion of parasite number due to each asexual stage replacing itself with multiple progeny can

generate the illusion of strong synchrony [33]. We also observed different degrees of synchrony across infections in which parasites became or remained synchronous. This could be due to a combined influence of multiple host rhythms on the IDC schedule, including minor contributions from non-feeding rhythms. For example, the decoupling of SCN-driven and peripheral rhythms in the WT TRF mice could impose conflicting timing information on the IDC compared to parasites in WT ad libitum fed mice who experienced better-aligned host rhythms. Furthermore, such conflict would not occur in the TRF clock-disrupted mice, in which only food-associated rhythms are present.

Why should a rhythm(s) associated with host feeding set the schedule for the parasite IDC? Food digestion provides glucose, for example, to the host and parasite, and blood glucose concentration follows a daily rhythm in hosts mounting immune responses [18]. Glucose tolerance changes during the day in a circadian manner and behavioural factors, such as host activity, feeding and fasting, strongly affect glucose metabolism. However, glucose regulation is a complex and tightly controlled process, achieved via the antagonistic effects of the hormones insulin and glucagon, and involves the contribution of several different organs (liver, pancreas) to dampen perturbations due to feeding and fasting. This makes it difficult to separate the contributions of host TTFL-clock-dependent and -independent processes on daily rhythms in glucose concentration [34]. In addition to glucose, IDC completion relies on other nutrients from the host's food, including amino acids essential for protein synthesis [35], purines (in particular hypoxanthine) for nucleic acid synthesis and lysophosphatidylcholine, for various processes such as cell membrane production [36]. Metabolomics-around-the-clock studies may help determine which rhythm(s) related to host feeding influences the IDC schedule.

Our results suggest that a product of food digestion schedules the IDC, supporting those of Prior et al. [19] and Hirako et al. [18], yet—at first glance—apparently contradicting two experiments, relating to food intake and infection of TTFL-clock disrupted mice, respectively, in Rijo-Ferreira et al. [10]. First, the food intake experiment in Rijo-Ferreira et al. [10] aimed to test if the act of eating itself schedules the IDC. They infected WT mice housed in LD cycles, thus despite food provision being spread evenly throughout the day and night, these hosts retained their nocturnal lifestyle, including undergoing the bulk of metabolism at night. In keeping with nocturnal rhythms, hosts whose food was spread evenly throughout the day and night seem to eat more pellets at night (higher night-time versus day-time mean in fig. 3C in Rijo-Ferreira et al. [10]). Second, like our design, Rijo-Ferreira et al. [10] gave TTFL-clock disrupted mice (Cry1/Cry2 null) food ad libitum and housed them in constant darkness, yet they found IDC rhythms (in infections started with synchronous parasites) remained strong. Rijo-Ferreira et al.'s [10] mice were kept in LD cycles until the point of infection which may allow residual rhythms generated by masking to persist for the first few days of infection. Indeed, when these TTFL-clock disrupted hosts are housed in constant darkness for a week before infection, IDC rhythms do become dampened (fig. 4L in Rijo-Ferreira et al.) [10].

While our experiments rule out a role for host TTFL-clock-driven rhythms in the IDC schedule, many host processes are rhythmic in clock-disrupted mice. For example, liver genes in clock-disrupted mice express rhythmicity

simply as a result of TRF protocols [37–40]. It is unclear to what extent this is due to a host endogenous oscillator independent of canonical TTFL-clock genes, such as that driving food anticipatory activity (FAA) [41–43]. Our study was not designed to quantify FAA, but nonetheless, our TRF fed clock-disrupted mice do exhibit behaviour consistent with FAA. Specifically, we observe a rise in body temperature and activity in anticipation of the 09.00 GMT feeding events (1–2 h before feeding; electronic supplementary material, figure S3). The precise mechanisms underpinning FAA rhythms are not fully understood, but they are thought to be independent of light-entrained oscillators and may use inputs such as levels of the 'hunger hormone' ghrelin, or insulin for entrainment [44]. Thus, it remains possible that the IDC schedule is aligned to the downstream consequences of such an oscillator. Similarly, daily oxidation–reduction rhythms exist within mammalian RBCs independent of a TTFL clock [45], are evolutionarily conserved [46] and may be linked to cellular flux in magnesium ions [47]. Recent work suggests that these rhythms are unlikely to impact on development during the IDC [9,24], but this is yet to be formally tested.

For small mammals, body temperature rhythms are influenced by a combination of the TTFL-clock, metabolism and locomotor activity. Almost all mice in which the IDC became or remained highly synchronous exhibited temperature rhythms to some extent (electronic supplementary material, figures S1–S3). Temperature rhythms can entrain host cells (including RBCs) and other parasites (e.g. Trypanosoma brucei) [48], and malaria parasites do respond to temperature change (e.g. to initiate gametogenesis when taken up by ectothermic mosquitoes) [49,50]. However, it is unlikely that the IDC schedule is aligned to a temperature rhythm. For example, Prior et al. [19] reveal inverted IDC rhythms in day- and night-fed mice, but host temperature rhythms are not inverted. More generally, temperature could only provide time-of-day resolution of 12 h and the IDC schedule is more precise than, for example, completion 'occurring at any point during the host's warm phase'. Alternatively, parasites may use a sharp change in temperature to determine time of day; however, multiple sharp temperature rises and drops exist throughout the day (electronic supplementary material, figure S3b) which suggest temperate change is also an unreliable indicator for time of day. A solution to this could be that only certain IDC stages are sensitive to temperature (so, misleading temperature changes are ignored), but if this were the case then parasites in infections mismatched to host rhythms (the WT and Per1/2-null TRF in both experiments presented here and in [11–13,19]) would not become rescheduled.

We also used our data to test whether parasite performance is enhanced in clock-disrupted hosts, potentially due to lack of regulation/coordination of TTFL-clock-controlled immune responses [25,28]. However, we find that the maximum parasite density is approximately 25–40% (across both experiments) lower in infections of clock-disrupted compared to WT hosts. Clock-disruption might reduce the ability of hosts to process and metabolize food efficiently, making these hosts a poorer resource for parasites. For example, PER1 and PER2 have a regulatory role in the circadian control of haem synthesis [51], with haemoglobin catabolism providing a primary source of amino acids for parasites, and loss of PER2 is implicated in making RBCs more susceptible to

oxidative stress, decreasing levels of ATP and shortening RBC lifespan [52]. Further, clock-disruption affects host nutrition via an interplay with microbiota [53]. Parasite performance is linked to host nutrition because caloric restriction leads to reduced parasite densities [54]. However, if either clock disruption and/or our time-restricted-feeding regime caused caloric restriction, we would expect this to manifest as clock-disrupted mice—especially in the TRF group—as having the lowest weights or the greatest weight loss. By contrast, clock-disrupted TRF mice were the heaviest in experiment 1 and clock-disrupted ad libitum fed mice lost the most weight in Experiment 2 (electronic supplementary material, table S5). Another, non-mutually exclusive, possibility is that the IDC becomes rescheduled faster in WT mice, and the faster that parasites can reschedule, the lower the fitness costs of being uncoordinated with the host's feeding rhythm. However, that parasite performance does not differ between infections remaining/becoming desynchronized versus synchronous within the same type of host (i.e. *Per1/2*-null) suggests either that there are no major costs to parasites of being desynchronized or that it is advantageous for them to match the degree of rhythmicity of their host. While the costs of virulence, as measured by weight loss, do not appear to differ between WT and clock-disrupted hosts, the findings for RBC loss are more complicated (and do not clearly mirror maximum parasite densities). No significant difference between feeding regimes or host genotypes was detected when infections were initiated with desynchronized parasites. But, in infections initiated with synchronous parasites, WT hosts became the most anaemic and clock-disrupted hosts with a feeding rhythm lost the fewest RBCs. Further work is needed to establish whether a loss of canonical clock regulation affects the ability of hosts to control or tolerate parasites.

## 5. Conclusion

The schedule (timing and synchrony) of the malaria parasite's IDC is not reliant on a functioning host core-TTFL clock. The speed with which the IDC schedule changes, its precision and the modest initial loss of parasite number involved in rescheduling [12] along with parasite control of IDC duration [24] suggest the parasite is actively aligning certain developmental stages with host feeding rhythms to take advantage of periodicity in a resource(s) it must acquire from the host's food processing. Recent studies suggest that IDC rhythms display a hallmark of a circadian clock [9,10], but other criteria (temperature compensation and entrainment) are yet to be met. Whatever the parasites' method of time-keeping, our data suggest it uses a signal stemming from the host's processing of food as a Zeitgeber or timing cue. Our data also highlight a complex interplay between host rhythms, features of the IDC schedule, parasite fitness (as approximated by maximum density) and disease severity. Unravelling these complexities may reveal interventions that minimize disease severity and improve recovery, while reducing parasite fitness.

Ethics. All procedures complied with the UK Animals (Scientific Procedures) Act 1986 (PPL 70/8546).

Data accessibility. Datasets are available in the Edinburgh DataShare repository (doi:10.7488/ds/2622).

Authors' contributions. A.J.O'D. and S.E.R. conceived the study, A.J.O'D. and K.F.P. carried out the experiments, A.J.O'D. analysed the data and all authors wrote the manuscript.

Competing interests. We declare we have no competing interests.

Funding. This work was supported by Wellcome (grant nos. 202769/Z/16/Z; 204511/Z/16/Z), the Royal Society (grant nos. UF110155; NF140517) and the Human Frontier Science Program (grant no. RGP0046/2013).

Acknowledgements. We thank Ronnie Mooney and Giles K. P. Barra for technical assistance.

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
