## [Reviewer comments · Proceedings of the Royal Society B: Biological Sciences]

Review History

RSPB-2019-2212.R0 (Original submission)

Review form: Reviewer 1

Recommendation

Reject – article is not of sufficient interest (we will consider a transfer to another journal)

Scientific importance: Is the manuscript an original and important contribution to its field?

Good

General interest: Is the paper of sufficient general interest?

Marginal

Quality of the paper: Is the overall quality of the paper suitable?

Good

Is the length of the paper justified?

Yes

Should the paper be seen by a specialist statistical reviewer?

No

Do you have any concerns about statistical analyses in this paper? If so, please specify them explicitly in your report.

No

It is a condition of publication that authors make their supporting data, code and materials available - either as supplementary material or hosted in an external repository. Please rate, if applicable, the supporting data on the following criteria.

Is it accessible?

Yes

Is it clear?

Yes

Is it adequate?

Yes

Do you have any ethical concerns with this paper?

No

Comments to the Author

I think the paper is well written and has great merit attributing to our understating of how parasites sychronize with their host. I do feel this paper is of a more narrow interest for those in the circadian field.

Review form: Reviewer 2

Recommendation

Major revision is needed (please make suggestions in comments)

Scientific importance: Is the manuscript an original and important contribution to its field?

Good

General interest: Is the paper of sufficient general interest?

Good

Quality of the paper: Is the overall quality of the paper suitable?

Marginal

Is the length of the paper justified?

Yes

Should the paper be seen by a specialist statistical reviewer?

No

Do you have any concerns about statistical analyses in this paper? If so, please specify them explicitly in your report.

No

It is a condition of publication that authors make their supporting data, code and materials available - either as supplementary material or hosted in an external repository. Please rate, if applicable, the supporting data on the following criteria.

Is it accessible?

Yes

Is it clear?

Yes

Is it adequate?

Yes

Do you have any ethical concerns with this paper?

No

Comments to the Author

In this manuscript by O'Donnell et al., the authors propose that rhythms in the malaria developmental cycle are determined by host feeding time and not the circadian clock of the host. The topic is extremely thought-provoking and the experiments in this study are interesting. However, some of the conclusions made by the authors are not supported by their data. There are also errors made in the interpretation or presentation of the circadian literature, some of the results are ambiguously represented, and some of the methods are lacking in details that may be critical to the interpretation of the data.

Major concerns:

a) Certain claims are not supported by the data. For example, in the abstract the authors state, "...whereas parasite rhythmicity was lost in clock-mutant mice that fed continuously". However, the results do not show the rhythms are lost, but instead that the amplitude is reduced. There is still a clear 24h rhythm, especially in Fig. 3.

This incorrect conclusion is repeated again on line 270, "and the IDC schedule is lost in hosts without a feeding rhythm." More generally, it is incorrect to say that there is no rhythm and then calculate the amplitude and phase of the IDC. The rhythm only shows reduced amplitude but is clearly there. This should be corrected throughout the manuscript (abstract, legend title, main text, etc).

b) The mouse behavior is unexpected. In particular, the WT (C57Bl/6J) behavior and temperature recordings prior to infection are not in line with published reports (DOI: <https://doi.org/10.1523/JNEUROSCI.10-11-03685>). Perhaps this is due to the very small number of mice per group? Or some other synchronizing factor? It has been well characterized that B6 mice would demonstrate a shorter period than 24h in DD. On the other hand, temperature rhythms are extremely short for those WT in TRF (~21h) which do not match the behavioral period rhythms or the 1 heatmap shown. Why is this? In addition, it is not expected that Per1/Per2 ko mice show no activity rhythms upon TRF, despite temperature rhythms, as there are studies that show entrainment upon TRF of arrhythmic mice (such as reference 40, doi:10.1371/journal.pone.0048892) yet there is no explanation of this.

c) The paragraph starting on line 305 is misleading and would benefit from being restated. It starts by stating that other rhythms other than food could be leading to the time cue. However, all the evidence given below this statement are examples that are consequences of feeding rhythms. So, in the first sentence it should be made clear that all the rhythms described below exist upon feeding rhythms being imposed.

Also, in line 307 the authors state, "For example, lipid levels in hepatic cells maintain oscillations in clock-disrupted mice (specifically Bmal1 knockouts) albeit in a different phase to wild type

mice [39].” Without clarification, this sentence should be removed as it leads to incorrect information. Indeed, there are rhythms observed in arrhythmic mutant mice, but these mice were kept in LD conditions (therefore with feeding & activity rhythms) and only released into DD on the day of collection (they were never arrhythmic). Therefore, this would be at the most comparable to Per1/Per2 TRF condition in the present study. Also, there was no Bmal1 KO in the particular study that they cite.

d) On line 318 the authors state, “However, it is unlikely that the IDC schedule is set by a temperature rhythm. Prior et al (2018) reveal inverted IDC rhythms in day- and night-fed mice but host temperature rhythms are not inverted.” However, in fig S1 there is an inversion of temperature upon daytime TRF, even in WT - minimum temperatures are now being recorded during nighttime, opposite of the WT ad lib (despite a burst at the beginning on the night potentially due to activity rhythms). If you plot timing of minimum temperature there would have been a very significant shift. This should be revised and explained.

e) In Figure 2, how was amplitude calculated? The values do not seem to match the plot on WT appears to have an amplitude > 0.5 . Is there confusion between the mathematical definition of amplitude with circadian amplitude?

f) On line 131, the authors state, “0.05 % para-aminobenzoic acid (to supplement parasite growth).” What is the diet composition? Why is this supplement used and did it lead to changes in host drinking patterns? Some diets modulate mouse eating patterns, controlling for the drinking behavior could influence the results since this nutrient has been associated with parasite growth. If there are changes in the host water ingestion, this would affect the cell-cycle, which is used as a readout of rhythms. (Also citations of studies that have used this before for this particular combination of mice strain and parasite should be included, or if not available the authors should characterize how it influences infection of this mouse and parasite combination, when supplementing with para-aminobenzoic acid).

g) On line 174, the authors state, “and whether locomotor activity can be used as a good proxy for feeding events (ESM).”. There should be data plotted from these video recording calculations since these were done and because it is an important aspect of the study.

h) On line 363, the authors state, “but we propose it is directly related to feeding events, and not associated with the food-entrained peripheral TTFL in the organs or the central oscillator in the SCN.” Feeding schedules are able to entrain peripheral oscillators TTFL, including in mutants. Even when the most upstream core clocks are mutated, downstream rhythms are observed as a consequence of these feedings. How are the authors distinguishing the direct effects of feeding and the effects that occur indirectly through the multiple host rhythms that arise as a consequence of feeding schedules?

Minor comments

Methods:

The authors should add the JAX or other animal supplier catalog number for mouse strains so that experiments can be replicated if desired.

There needs to be a better description of how they implemented daytime TRF? Did they add and remove food by hand with animals on the same cage, or instead did they change mice from a cage without food to a cage with food to avoid crumbs on the floor?

Decision letter (RSPB-2019-2212.R0)

25-Nov-2019

Dear Mr O'Donnell:

I am writing to inform you that your manuscript RSPB-2019-2212 entitled "Host circadian clocks do not set the schedule for the within-host replication of malaria parasites" has, in its current form, been rejected for publication in Proceedings B.

This action has been taken on the advice of referees, who have recommended that substantial and important revisions are necessary. With this in mind we would be happy to consider a resubmission, provided the comments of the referees are fully addressed. However please note that this is not a provisional acceptance.

Sincerely,
Professor Hans Heesterbeek
<mailto:proceedingsb@royalsociety.org>

Associate Editor
Board Member: 1
Comments to Author:

Two reviewers find considerable merit in this study. Reviewer 1 find the article to be sound, but too narrow for publication in PRSB. I disagree, but rather think the claims of this paper, if correct, will be of wide interest. Reviewer 2 finds the study compelling, but over-stated with some prior information mis-represented. The authors should revise their manuscript to satisfy the concerns of Reviewer 2.

Reviewer(s)' Comments to Author:

Referee: 1

Comments to the Author(s)

I think the paper is well written and has great merit attributing to our understating of how parasites synchronize with their host. I do feel this paper is of a more narrow interest for those in the circadian field.

Referee: 2

Comments to the Author(s)

In this manuscript by O'Donnell et al., the authors propose that rhythms in the malaria developmental cycle are determined by host feeding time and not the circadian clock of the host. The topic is extremely thought-provoking and the experiments in this study are interesting. However, some of the conclusions made by the authors are not supported by their data. There are also errors made in the interpretation or presentation of the circadian literature, some of the results are ambiguously represented, and some of the methods are lacking in details that may be critical to the interpretation of the data.

Major concerns:

a) Certain claims are not supported by the data. For example, in the abstract the authors state, "...whereas parasite rhythmicity was lost in clock-mutant mice that fed continuously". However, the results do not show the rhythms are lost, but instead that the amplitude is reduced. There is still a clear 24h rhythm, especially in Fig. 3.

This incorrect conclusion is repeated again on line 270, "and the IDC schedule is lost in hosts without a feeding rhythm." More generally, it is incorrect to say that there is no rhythm and then calculate the amplitude and phase of the IDC. The rhythm only shows reduced amplitude but is clearly there. This should be corrected throughout the manuscript (abstract, legend title, main text, etc).

b) The mouse behavior is unexpected. In particular, the WT (C57Bl/6J) behavior and temperature recordings prior to infection are not in line with published reports (DOI: <https://doi.org/10.1523/JNEUROSCI.10-11-03685>). Perhaps this is due to the very small number of mice per group? Or some other synchronizing factor? It has been well characterized that B6 mice would demonstrate a shorter period than 24h in DD. On the other hand, temperature rhythms are extremely short for those WT in TRF (~21h) which do not match the behavioral period rhythms or the 1 heatmap shown. Why is this? In addition, it is not expected that Per1/Per2 ko mice show no activity rhythms upon TRF, despite temperature rhythms, as there are studies that show entrainment upon TRF of arrhythmic mice (such as reference 40, doi:10.1371/journal.pone.0048892) yet there is no explanation of this.

c) The paragraph starting on line 305 is misleading and would benefit from being restated. It starts by stating that other rhythms other than food could be leading to the time cue. However, all the evidence given below this statement are examples that are consequences of feeding rhythms. So, in the first sentence it should be made clear that all the rhythms described below exist upon feeding rhythms being imposed.

Also, in line 307 the authors state, "For example, lipid levels in hepatic cells maintain oscillations in clock-disrupted mice (specifically Bmal1 knockouts) albeit in a different phase to wild type mice [39]." Without clarification, this sentence should be removed as it leads to incorrect information. Indeed, there are rhythms observed in arrhythmic mutant mice, but these mice were kept in LD conditions (therefore with feeding & activity rhythms) and only released into DD on the day of collection (they were never arrhythmic). Therefore, this would be at the most comparable to Per1/Per2 TRF condition in the present study. Also, there was no Bmal1 KO in the particular study that they cite.

- d) On line 318 the authors state, "However, it is unlikely that the IDC schedule is set by a temperature rhythm. Prior et al (2018) reveal inverted IDC rhythms in day- and night-fed mice but host temperature rhythms are not inverted." However, in fig S1 there is an inversion of temperature upon daytime TRF, even in WT - minimum temperatures are now being recorded during nighttime, opposite of the WT ad lib (despite a burst at the beginning on the night potentially due to activity rhythms). If you plot timing of minimum temperature there would have been a very significant shift. This should be revised and explained.
- e) In Figure 2, how was amplitude calculated? The values do not seem to match the plot on WT appears to have an amplitude > 0.5. Is there confusion between the mathematical definition of amplitude with circadian amplitude?
- f) On line 131, the authors state, "0.05 % para-aminobenzoic acid (to supplement parasite growth)." What is the diet composition? Why is this supplement used and did it lead to changes in host drinking patterns? Some diets modulate mouse eating patterns, controlling for the drinking behavior could influence the results since this nutrient has been associated with parasite growth. If there are changes in the host water ingestion, this would affect the cell-cycle, which is used as a readout of rhythms. (Also citations of studies that have used this before for this particular combination of mice strain and parasite should be included, or if not available the authors should characterize how it influences infection of this mouse and parasite combination, when supplementing with para-aminobenzoic acid).
- g) On line 174, the authors state, "and whether locomotor activity can be used as a good proxy for feeding events (ESM)". There should be data plotted from these video recording calculations since these were done and because it is an important aspect of the study.
- h) On line 363, the authors state, "but we propose it is directly related to feeding events, and not associated with the food-entrained peripheral TTFL in the organs or the central oscillator in the SCN." Feeding schedules are able to entrain peripheral oscillators TTFL, including in mutants. Even when the most upstream core clocks are mutated, downstream rhythms are observed as a consequence of these feedings. How are the authors distinguishing the direct effects of feeding and the effects that occur indirectly through the multiple host rhythms that arise as a consequence of feeding schedules?

Minor comments

Methods:

The authors should add the JAX or other animal supplier catalog number for mouse strains so that experiments can be replicated if desired.

There needs to be a better description of how they implemented daytime TRF? Did they add and remove food by hand with animals on the same cage, or instead did they change mice from a cage without food to a cage with food to avoid crumbs on the floor?

Author's Response to Decision Letter for (RSPB-2019-2212.R0)

See Appendix A.

RSPB-2020-0347.R0

Review form: Reviewer 2

Recommendation

Accept with minor revision (please list in comments)

Scientific importance: Is the manuscript an original and important contribution to its field?

Acceptable

General interest: Is the paper of sufficient general interest?

Acceptable

Quality of the paper: Is the overall quality of the paper suitable?

Acceptable

Is the length of the paper justified?

No

Should the paper be seen by a specialist statistical reviewer?

No

Do you have any concerns about statistical analyses in this paper? If so, please specify them explicitly in your report.

No

It is a condition of publication that authors make their supporting data, code and materials available - either as supplementary material or hosted in an external repository. Please rate, if applicable, the supporting data on the following criteria.

Is it accessible?

N/A

Is it clear?

N/A

Is it adequate?

N/A

Do you have any ethical concerns with this paper?

No

Comments to the Author

In this revised version of the manuscript by Odonnell et al., the authors made significant changes that in our view have greatly improved the scientific accuracy and support of their conclusions. Except for some minor points highlighted below, all other concerns have been addressed.

1) Line 295: "food anticipatory activity (FAA; which requires an endogenous oscillator)" This needs to be rephrased to clarify to which oscillator they are referring. SCN-ablated animals still exhibit normal FAA, indicating that the food-entrainable oscillator is located outside of the SCN (Mistlberger, 1994). It does not seem to rely on the canonical molecular circadian timekeeping mechanism. However, without clarifying which oscillator is meant, the authors contradict themselves in the same sentence:

“food anticipatory activity (FAA; which requires an endogenous oscillator) can be observed in clock-disrupted mice [40]”

2) The authors refer to Food Anticipatory activity in arrhythmic mutant mice upon restricted feeding, as an example. It would be good to clarify that 12h time restriction has also been shown to rescue rhythmicity in arrhythmic mice, e.g. gene expression rhythmicity upon time restricted feeding in mutant mice (doi.org/10.1073/pnas.1515308112). The authors should read the circadian literature and include additional relevant citations. There are studies from the Schibler lab and Panda lab, among others, that have shown this. Thus, it is not surprising that feeding rhythms can entrain the parasite population since they are entraining host rhythms, and host rhythms (as a whole without knowing exactly which cues) have been shown to entrain parasite population rhythms. This is now acknowledged by the authors, but the manuscript would benefit with them clarifying that this exact protocol was expected to show the observed results.

3) We appreciate that the authors attempted to verify that their assessment of activity was a good proxy for food intake. However, we feel this still requires further work:

a) There is no axis information for Fig. S4

b) Mice are group housed, so is this information from a single mouse? More detailed information is needed about what is represented in the figure.

c) We do not believe this is the best analysis/representation because there is no y axis. It seems that a small, 1 min movement would count the same as a 15 min active bout, which leads to the impression that mice are hyperactive. Perhaps a heatmap of some type, or simply including a clearly labelled y axis would avoid confusion.

4) Please add details in each figure legend about the day of infection measured, and how many mice are represented in each figure for easier understanding without having to refer to the manuscript.

Review form: Reviewer 3

Recommendation

Reject – article is scientifically unsound

Scientific importance: Is the manuscript an original and important contribution to its field?

Acceptable

General interest: Is the paper of sufficient general interest?

Acceptable

Quality of the paper: Is the overall quality of the paper suitable?

Poor

Is the length of the paper justified?

Yes

Should the paper be seen by a specialist statistical reviewer?

No

Do you have any concerns about statistical analyses in this paper? If so, please specify them explicitly in your report.

Yes

It is a condition of publication that authors make their supporting data, code and materials available - either as supplementary material or hosted in an external repository. Please rate, if applicable, the supporting data on the following criteria.

Is it accessible?

Yes

Is it clear?

No

Is it adequate?

No

Do you have any ethical concerns with this paper?

No

Comments to the Author

Summary: The IDC character is not properly validated with a sufficient number of time points. The key is to increase the number of time points for IDC. An analysis with JTK_CYCLE is needed to provide a nonparametric analysis of circadian rhythms because of the small number of time points (9) on the IDC character. There are no molecular correlates (such as RNAs followed or proteins followed) to support the hypothesis of independent circadian rhythms in parasite. A proper phase, period, and amplitude analysis is not given. For example, what is the period? A more thorough analysis of phase is needed. See, for example, Caranica et al. (2019) in *Yale Biology and Medicine* 92: 169. Even given that IDC is rhythmic under ad libitum feeding and the absence of a host clock, how can the authors exclude other host queues, such as body temperature? While the L/D cycle of the host was eliminated, what was done about entrainment to temperature? What record of cage temperatures was maintained? What are the RBC levels doing in the infected mice? Does this plasmodium generate recrudescence infections?

Specific Comments (L is line number)

L41. No molecular correlates are presented in the parasite.

L136. What happens with different strains of plasmodium?

L168. Two complete cycles is needed at a minimum. Sample size is too small with 9 time points. In general there needs to be more time points and more cycles.

L191. There is a need to report phase, period, and amplitude with standard errors. There is a need to report percentage variation explained by each COSINOR analysis. If $A\cos(\omega t + \phi) + \text{error}$ is being fitted, say so. Also what is the percentage variation explained by this simple model when the nonlinear regression on time is done?

L207. What is RBC doing over time in the mice? Does this plasmodium cause a recrudescence infection, and what is its period? Some recrudescence infections in primate species and with particular plasmodium species can cause recrudescence infections with a period of 24 h.

L208. With only 9 time points a nonparametric method is needed, such as JTK_CYCLE.

L255. To reach the conclusion that IDC has a clock additional experiments are needed. It must be shown that IDC not only has a period of 24 h (which is not reported in this paper), but that IDC is subject to Zeitgeber entrainment and temperature compensation. What about these other properties? Does the IDC respond to a different light schedule or food schedule for the host?

Decision letter (RSPB-2020-0347.R0)

06-Apr-2020

Dear Mr O'Donnell:

Your manuscript has now been peer reviewed and the reviews have been assessed by an Associate Editor. The reviewers' comments (not including confidential comments to the Editor) and the comments from the Associate Editor are included at the end of this email for your reference. As you will see, the reviewers differ in their opinion. One reviewer also assessed the previous version of the manuscript and feels the manuscript has improved. That reviewer is satisfied, apart from a number of (relatively) minor issues. The dilemma is posed by the other reviewer (3), who is new to this version of the manuscript. That reviewer has substantial and profound doubts and recommends rejection. The Associate Editor and I feel that, because this reviewer is new, you should at least have an opportunity to respond to the new criticism. Therefore, we would like to invite you to revise your manuscript to see whether you can address the issues in a satisfactory way.

Research ethics:

Use of animals and field studies:

Please submit a copy of your revised paper within three weeks. If we do not hear from you within this time your manuscript will be rejected. If you are unable to meet this deadline please let us know as soon as possible, as we may be able to grant a short extension.

Best wishes,
Professor Hans Heesterbeek
mailto: proceedingsb@royalsociety.org

Associate Editor Board Member

Comments to Author:

Two expert referees reviewed this manuscript. One reviewer has profound problems. Under these conditions PRSB cannot accept the work for publication in its present form.

Reviewer(s)' Comments to Author:

Referee: 3

Comments to the Author(s).

Summary: The IDC character is not properly validated with a sufficient number of time points. The key is to increase the number of time points for IDC. An analysis with JTK_CYCLE is needed to provide a nonparametric analysis of circadian rhythms because of the small number of time points (9) on the IDC character. There are no molecular correlates (such as RNAs followed or proteins followed) to support the hypothesis of independent circadian rhythms in parasite. A proper phase, period, and amplitude analysis is not given. For example, what is the period? A more thorough analysis of phase is needed. See, for example, Caranica et al. (2019) in *Yale Biology and Medicine* 92: 169. Even given that IDC is rhythmic under ad libitum feeding and the absence of a host clock, how can the authors exclude other host cues, such as body temperature? While the L/D cycle of the host was eliminated, what was done about entrainment to temperature? What record of cage temperatures was maintained? What are the RBC levels doing in the infected mice? Does this plasmodium generate recrudescence infections?

Specific Comments (L is line number)

L41. No molecular correlates are presented in the parasite.

L136. What happens with different strains of plasmodium?

L168. Two complete cycles is needed at a minimum. Sample size is too small with 9 time points. In general there needs to be more time points and more cycles.

L191. There is a need to report phase, period, and amplitude with standard errors. There is a need to report percentage variation explained by each COSINOR analysis. If $\text{Acos}(\omega t + \phi) + \text{error}$ is being fitted, say so. Also what is the percentage variation explained by this simple model when the nonlinear regression on time is done?

L207. What is RBC doing over time in the mice? Does this plasmodium cause a recrudescence infection, and what is its period? Some recrudescence infections in primate species and with particular plasmodium species can cause recrudescence infections with a period of 24 h.

L208. With only 9 time points a nonparametric method is needed, such as JTK_CYCLE.

L255. To reach the conclusion that IDC has a clock additional experiments are needed. It must be shown that IDC not only has a period of 24 h (which is not reported in this paper), but that IDC is subject to Zeitgeber entrainment and temperature compensation. What about these other properties? Does the IDC respond to a different light schedule or food schedule for the host?

Referee: 2

Comments to the Author(s).

In this revised version of the manuscript by Odonnell et al., the authors made significant changes that in our view have greatly improved the scientific accuracy and support of their conclusions. Except for some minor points highlighted below, all other concerns have been addressed.

1) Line 295: "food anticipatory activity (FAA; which requires an endogenous oscillator)" This needs to be rephrased to clarify to which oscillator they are referring. SCN-ablated animals still exhibit normal FAA, indicating that the food-entrainable oscillator is located outside of the SCN (Mistlberger, 1994). It does not seem to rely on the canonical molecular circadian

timekeeping mechanism. However, without clarifying which oscillator is meant, the authors contradict themselves in the same sentence:

“food anticipatory activity (FAA; which requires an endogenous oscillator) can be observed in clock-disrupted mice [40]”

2) The authors refer to Food Anticipatory activity in arrhythmic mutant mice upon restricted feeding, as an example. It would be good to clarify that 12h time restriction has also been shown to rescue rhythmicity in arrhythmic mice, e.g. gene expression rhythmicity upon time restricted feeding in mutant mice (doi.org/10.1073/pnas.1515308112). The authors should read the circadian literature and include additional relevant citations. There are studies from the Schibler lab and Panda lab, among others, that have shown this. Thus, it is not surprising that feeding rhythms can entrain the parasite population since they are entraining host rhythms, and host rhythms (as a whole without knowing exactly which cues) have been shown to entrain parasite population rhythms. This is now acknowledged by the authors, but the manuscript would benefit with them clarifying that this exact protocol was expected to show the observed results.

3) We appreciate that the authors attempted to verify that their assessment of activity was a good proxy for food intake. However, we feel this still requires further work:

a) There is no axis information for Fig. S4

b) Mice are group housed, so is this information from a single mouse? More detailed information is needed about what is represented in the figure.

c) We do not believe this is the best analysis/representation because there is no y axis. It seems that a small, 1 min movement would count the same as a 15 min active bout, which leads to the impression that mice are hyperactive. Perhaps a heatmap of some type, or simply including a clearly labelled y axis would avoid confusion.

4) Please add details in each figure legend about the day of infection measured, and how many mice are represented in each figure for easier understanding without having to refer to the manuscript.

Author's Response to Decision Letter for (RSPB-2020-0347.R0)

See Appendix B.

RSPB-2020-0347.R1 (Revision)

Review form: Reviewer 2

Recommendation

Accept as is

Scientific importance: Is the manuscript an original and important contribution to its field?

Good

General interest: Is the paper of sufficient general interest?

Good

Quality of the paper: Is the overall quality of the paper suitable?

Good

Is the length of the paper justified?

Yes

Should the paper be seen by a specialist statistical reviewer?

No

Do you have any concerns about statistical analyses in this paper? If so, please specify them explicitly in your report.

No

It is a condition of publication that authors make their supporting data, code and materials available - either as supplementary material or hosted in an external repository. Please rate, if applicable, the supporting data on the following criteria.

Is it accessible?

Yes

Is it clear?

Yes

Is it adequate?

Yes

Do you have any ethical concerns with this paper?

No

Comments to the Author

In this revised version of the manuscript by Odonnell et al., the authors have addressed our concerns.

Review form: Reviewer 3

Recommendation

Reject - article is scientifically unsound

Scientific importance: Is the manuscript an original and important contribution to its field?

Good

General interest: Is the paper of sufficient general interest?

Good

Quality of the paper: Is the overall quality of the paper suitable?

Marginal

Is the length of the paper justified?

Yes

Should the paper be seen by a specialist statistical reviewer?

No

Do you have any concerns about statistical analyses in this paper? If so, please specify them explicitly in your report.

Yes

It is a condition of publication that authors make their supporting data, code and materials available - either as supplementary material or hosted in an external repository. Please rate, if applicable, the supporting data on the following criteria.

Is it accessible?

N/A

Is it clear?

N/A

Is it adequate?

N/A

Do you have any ethical concerns with this paper?

No

Comments to the Author

Comments to the author are attached as a file. (See Appendix C)

Decision letter (RSPB-2020-0347.R1)

26-May-2020

Dear Mr O'Donnell:

Your manuscript has now been peer reviewed and the reviews have been assessed by an Associate Editor. The reviewers' comments (not including confidential comments to the Editor) and the comments from the Associate Editor are included at the end of this email for your reference. As you will see, the reviewers have raised some issues with your manuscript and we would like to invite you to revise your manuscript to address them.

As a rule, we do not allow multiple rounds of revision so we urge you to make every effort to fully address all of the comments at this stage. I appreciate your concern over the speed with which your field is currently evolving, with several recent publications you mention in high-impact journals. I hope you can, however, also appreciate that manuscripts have to be judged on their merit and that the critical reviewer makes a number of points that will need to be addressed. If deemed necessary by the Associate Editor, your manuscript will be sent back to one or more of the original reviewers for assessment. If the original reviewers are not available we may invite new reviewers. Please note that we cannot guarantee eventual acceptance of your manuscript at this stage.

When submitting your revision please upload a file under "Response to Referees" in the "File Upload" section. This should document, point by point, how you have responded to the reviewers' and Editors' comments, and the adjustments you have made to the manuscript. We

require a copy of the manuscript with revisions made since the previous version marked as 'tracked changes' to be included in the 'response to referees' document.

Research ethics:

Use of animals and field studies:

Please submit a copy of your revised paper within three weeks. If we do not hear from you within this time your manuscript will be rejected. If you are unable to meet this deadline please let us know as soon as possible, as we may be able to grant a short extension.

Best wishes,
Professor Hans Heesterbeek
Editor, Proceedings B
mailto: proceedingsb@royalsociety.org

Associate Editor
Board Member: 1
Comments to Author:
Reviewer 3 has made additional comments which would need to be satisfactorily addressed before this paper could be accepted in PRSB

Reviewer(s)' Comments to Author:

Referee: 3

Comments to the Author(s)
Comments to the author are attached as a file.

Referee: 2

Comments to the Author(s)
In this revised version of the manuscript by Odonnell et al., the authors have addressed our concerns.

Author's Response to Decision Letter for (RSPB-2020-0347.R1)

See Appendix D.

RSPB-2020-0347.R2 (Revision)

Review form: Reviewer 3

Recommendation

Accept as is

Scientific importance: Is the manuscript an original and important contribution to its field?

Good

General interest: Is the paper of sufficient general interest?

Good

Quality of the paper: Is the overall quality of the paper suitable?

Good

Is the length of the paper justified?

Yes

Should the paper be seen by a specialist statistical reviewer?

No

Do you have any concerns about statistical analyses in this paper? If so, please specify them explicitly in your report.

No

It is a condition of publication that authors make their supporting data, code and materials available - either as supplementary material or hosted in an external repository. Please rate, if applicable, the supporting data on the following criteria.

Is it accessible?

Yes

Is it clear?

Yes

Is it adequate?

Yes

Do you have any ethical concerns with this paper?

No

Comments to the Author

Comments are attached. (See Appendix E)

Decision letter (RSPB-2020-0347.R2)

13-Jul-2020

Dear Mr O'Donnell

I am pleased to inform you that your manuscript RSPB-2020-0347.R2 entitled "Host circadian clocks do not set the schedule for the within-host replication of malaria parasites" has been accepted for publication in Proceedings B.

The remaining referee has recommended publication, but also responds to your rebuttal to explain his/her reasoning. Because this clarification may lead to additional minor changes, I give you the opportunity to respond to the referee's comments and revise your manuscript one final time. Because the schedule for publication is very tight, it is a condition of publication that you submit the revised version of your manuscript within 7 days. If you do not think you will be able to meet this date please let us know.

To revise your manuscript, log into <https://mc.manuscriptcentral.com/prsb> and enter your Author Centre, where you will find your manuscript title listed under "Manuscripts with

Decisions." Under "Actions," click on "Create a Revision." Your manuscript number has been appended to denote a revision. You will be unable to make your revisions on the originally submitted version of the manuscript. Instead, revise your manuscript and upload a new version through your Author Centre.

[http://datadryad.org/submit?journalID=RSPB&manu=\(Document not available\)](http://datadryad.org/submit?journalID=RSPB&manu=(Document+not+available)) which will take you to your unique entry in the Dryad repository. If you have already submitted your data to dryad you can make any necessary revisions to your dataset by following the above link.

Please see <https://royalsociety.org/journals/ethics-policies/data-sharing-mining/> for more details.

Sincerely,

Professor Hans Heesterbeek
Editor, Proceedings B
<mailto:proceedingsb@royalsociety.org>

Reviewer(s)' Comments to Author:

Referee: 3

Comments to the Author(s)
Comments are attached.

Author's Response to Decision Letter for (RSPB-2020-0347.R2)

See Appendix F.

Decision letter (RSPB-2020-0347.R3)

21-Jul-2020

Dear Mr O'Donnell

I am pleased to inform you that your manuscript entitled "Host circadian clocks do not set the schedule for the within-host replication of malaria parasites" has been accepted for publication in Proceedings B.

Open Access

Paper charges

Sincerely,

Appendix A

Dear Professor Heesterbeek

Many thanks for sending our manuscript out to review and giving us the opportunity to revise it (RSPB-2019-2212). We are grateful to the referees for their insight and constructive remarks and are very pleased both the Reviewers and Associate Editor feel the manuscript is thought-provoking and of interest to a wider audience.

We have revised our manuscript according to the reviewers' comments and suggestions, as outlined below. The word count of the revised manuscript is 7,213.

Sincerely,

Aidan O'Donnell, Kimberley Prior and Sarah Reece

Referee: 1

I think the paper is well written and has great merit attributing to our understating of how parasites synchronize with their host. I do feel this paper is of a more narrow interest for those in the circadian field.

Many thanks for the positive comments on the paper. Regarding its level of interest, we feel that the merger of evolutionary ecology with chronobiology and a medically important disease has sufficient impact for PRSLb. For instance, recent papers on the topic have proved to be of high interest beyond the field of chronobiology, being published in top parasitology journals [1, 2], and our lab has just published an invited review in Cell, Host, and Microbe [3].

Referee: 2

In this manuscript by O'Donnell et al., the authors propose that rhythms in the malaria developmental cycle are determined by host feeding time and not the circadian clock of the host. The topic is extremely thought-provoking and the experiments in this study are interesting. However, some of the conclusions made by the authors are not supported by their data. There are also errors made in the interpretation or presentation of the circadian literature, some of the results are ambiguously represented, and some of the methods are lacking in details that may be critical to the interpretation of the data.

Major concerns:

- a) ***Certain claims are not supported by the data. For example, in the abstract the authors state, "...whereas parasite rhythmicity was lost in clock-mutant mice that fed continuously". However, the results do not show the rhythms are lost, but instead that the amplitude is reduced. There is still a clear 24h rhythm, especially in Fig. 3. This incorrect conclusion is repeated again on line 270, "and the IDC schedule is lost in hosts without a feeding rhythm." More generally, it is incorrect to say that there is no rhythm and then calculate the amplitude and phase of the IDC. The rhythm only shows reduced amplitude but is clearly there. This should be corrected throughout the manuscript (abstract, legend title, main text, etc).***

We agree with the reviewer that our data reveal that parasites lose some – but not all – rhythmicity in hosts without feeding rhythms. Our expectation is that we have taken a snapshot of infections during

the process of losing rhythmicity and so, their amplitude would have been further dampened in subsequent IDCs. However, the reviewer is correct, that we don't demonstrate a complete loss of rhythmicity, and so we have rephrased all statements about the loss of rhythmicity to be more accurate and explained our premise more clearly.

b) *The mouse behavior is unexpected. In particular, the WT (C57Bl/6J) behavior and temperature recordings prior to infection are not in line with published reports (DOI: <https://doi.org/10.1523/JNEUROSCI.10-11-03685>). Perhaps this is due to the very small number of mice per group? Or some other synchronizing factor? It has been well characterized that B6 mice would demonstrate a shorter period than 24h in DD.*

We agree that the periods observed in our WT mice are on average 0.2h longer than other published reports despite our mice being subjected to free-running conditions (DD, with constant dim red led light). There are some differences in methodology in our work including using RFID tags to measure all locomotor activity rather than voluntary wheel running behaviour alone, which would not include foraging activity for example. Perhaps this could account for the difference? However, the actual FRP is not integral for the findings of the manuscript. For example, if an unknown synchronizing factor was present and more important to parasites than a feeding rhythm, we would expect to observe the same parasite rhythms in all treatment groups (but we observe that parasite rhythms match host feeding rhythms). Rather, free-running conditions used as a tool to create the arrhythmia in the *Per1/2*-null mutant mice (as per [4]). However, we have clarified that unlike other studies, we monitor all activity, and thus have rewritten the results and ESM to report observed periods rather than declare them truly free-running.

On the other hand, temperature rhythms are extremely short for those WT in TRF (~21h) which do not match the behavioral period rhythms or the 1 heatmap shown. Why is this?

We thank the reviewer for highlighting the error in the reported temperature periods in the ESM text and those predicted from Figure 1. This was a coding error and has since been corrected in the text, which is now consistent with Figure 1.

In addition, it is not expected that *Per1/Per2* ko mice show no activity rhythms upon TRF, despite temperature rhythms, as there are studies that show entrainment upon TRF of arrhythmic mice (such as reference 40, doi:10.1371/journal.pone.0048892) yet there is no explanation of this.

We expect the reviewer is referring to Food Anticipatory Activity (FAA)? FAA has indeed been demonstrated in many knockout mice models [5-8]. We note that many of these studies employed a shorter window of food availability than our study (we were conservative in our approach because of the energetic costs of malaria infection). However, our data also suggest a possible FAA rhythm because mice exhibit increases in activity and temperature 1-2 hours before the 9am feeding time. We have made additional figures for the ESM (Figures S3a & b) to better illustrate the average locomotor activity and temperature rhythms throughout 24hrs to highlight the potential for FAA, and expanded our discussion to include this phenomenon and its potential role in parasites rhythms (Lines 291-307).

- c) ***The paragraph starting on line 305 is misleading and would benefit from being restated. It starts by stating that other rhythms other than food could be leading to the time cue. However, all the evidence given below this statement are examples that are consequences of feeding rhythms. So, in the first sentence it should be made clear that all the rhythms described below exist upon feeding rhythms being imposed. Also, in line 307 the authors state, “For example, lipid levels in hepatic cells maintain oscillations in clock-disrupted mice (specifically Bmal1 knockouts) albeit in a different phase to wild type mice [39].” Without clarification, this sentence should be removed as it leads to incorrect information. Indeed, there are rhythms observed in arrhythmic mutant mice, but these mice were kept in LD conditions (therefore with feeding & activity rhythms) and only released into DD on the day of collection (they were never arrhythmic). Therefore, this would be at the most comparable to Per1/Per2 TRF condition in the present study. Also, there was no Bmal1 KO in the particular study that they cite.***

We apologise for the interpretation of reference 39 and thank for the reviewer for highlighting our error. We agree that these rhythms are dependent on feeding rhythms and have re-written the discussion to take this into account.

- d) ***On line 318 the authors state, “However, it is unlikely that the IDC schedule is set by a temperature rhythm. Prior et al (2018) reveal inverted IDC rhythms in day- and night-fed mice but host temperature rhythms are not inverted.” However, in fig S1 there is an inversion of temperature upon daytime TRF, even in WT – minimum temperatures are now being recorded during nighttime, opposite of the WT ad lib (despite a burst at the beginning on the night potentially due to activity rhythms). If you plot timing of minimum temperature there would have been a very significant shift. This should be revised and explained.***

This is an interesting discussion point. We agree that we do see temperature rhythms in our TRF mice close that is close to a full inversion when compared to WT mice (ESM Figure S3b). This differs from Prior *et al* in which no complete inversion was seen, perhaps due the differences in methodology e.g. mice strain (Prior used MF1 vs our C57/bl6), housing (Prior group housed animals while we housed mice individually) and differences in TRF timing (Prior provided food for 12 hours vs our 10 hours). However, it is well known that malaria parasites have an ability to respond to temperature changes (gametogenesis in transmission stages is partly triggered by temperature) [9]. Yet, despite these points, we doubt host temperature drives parasite rhythms because responding to absolute temperature in the mammalian host would only give parasites a 12hour resolution to their timekeeping (i.e. cold in the day vs warm at night), and thus we would expect to see lower stage amplitudes. Alternatively, parasites may respond to a sharp change in temperature (either using an increase or decrease in host temperature as a timing cue). This creates two scenarios, both of which we suspect are unlikely:

- 1) **Parasites are scheduled by a sharp increase in temperature:** This is unlikely to be a reliable signal to parasites as there are several occasions in the 24hr period when temperature increases

occur. For example, in *ad lib* fed WT mice, which are more analogous to natural conditions than the other groups, (ESM Figure S3b) temperature rises sharply in the early part of the light period and again early in the dark period. Thus, if parasites were scheduled solely by an increase in temperature, we would expect infections to stochastically follow the light or dark increases in temperature, resulting in 50% of infections in this treatment group having each, opposing, schedule. Instead we observe very repeatable parasite rhythms within in this group.

- 2) **Parasites are scheduled by a sharp decrease in temperature:** Again, this is unlikely to be a reliable signal. In *ad lib* WT mice, there are two drops in temperature spaced 3 hours apart during the light period, and in TRF WT mice there are two significant drops in temperature spaced 12 hours apart (early in the light period and early in the dark period; ESM Figure S3b).

It is theoretically possible that parasites “ignore” a cue when they are at certain stages in the IDC, which enables them to respond to the “correct” cue at the right time, but this is not consistent with observations. For example, this strategy cannot explain the rescheduling of parasites that have been mismatched by 12 hours to the host’s rhythm, because the parasites wouldn’t know that the cue appearing in time with their IDC stage is incorrect and thus never reschedule ([2, 10-12] and this study in which parasites in TRF mice were initiated out of phase with the feeding rhythm).

We do recognize that our experiments were not specifically designed to dissect the parasites ability to align their rhythms with temperature changes in host, but our observations suggest Occam’s Razor is not supportive of temperature scheduling parasites. However, we have revised the discussion section to include the potential effects of temperature on parasite rhythms (Lines 308-323)

- e) ***In Figure 2, how was amplitude calculated? The values do not seem to match the plot on WT appears to have an amplitude > 0.5. Is there confusion between the mathematical definition of amplitude with circadian amplitude?***

Amplitude was calculated using the formula $\sqrt{a^2+b^2}$ to give mathematical amplitude and then multiplied by two for circadian amplitude. In Figure 2, mathematical amplitude was reported in erratum. We have altered the figure to report circadian amplitude and maintain consistency throughout the manuscript.

- f) ***On line 131, the authors state, “0.05 % para-aminobenzoic acid (to supplement parasite growth).” What is the diet composition? Why is this supplement used and did it lead to changes in host drinking patterns? Some diets modulate mouse eating patterns, controlling for the drinking behavior could influence the results since this nutrient has been associated with parasite growth. If there are changes in the host water ingestion, this would affect the cell-cycle, which is used as a readout of rhythms. (Also citations of studies that have used this before for this particular combination of mice strain and parasite should be included, or if not available the authors should characterize how it***

influences infection of this mouse and parasite combination, when supplementing with para-aminobenzoic acid).

The addition of PABA in drinking water is standard procedure in experimental malaria infections [13-18] because the parasites require folate from the host. However, because mice – in all treatment groups in both experiments - had *ad lib* access to PABA-drinking-water, but we observe different parasite rhythms across treatment groups suggests that our food rhythm perturbations by far the greatest effect on parasite rhythms.

g) *On line 174, the authors state, “and whether locomotor activity can be used as a good proxy for feeding events (ESM)”. There should be data plotted from these video recording calculations since these were done and because it is an important aspect of the study.*

We have added ESM Figure S4 to illustrate the correlation between the timing of feeding events and locomotor activity.

h) *On line 363, the authors state, “but we propose it is directly related to feeding events, and not associated with the food-entrained peripheral TTFL in the organs or the central oscillator in the SCN.” Feeding schedules are able to entrain peripheral oscillators TTFL, including in mutants. Even when the most upstream core clocks are mutated, downstream rhythms are observed as a consequence of these feedings. How are the authors distinguishing the direct effects of feeding and the effects that occur indirectly through the multiple host rhythms that arise as a consequence of feeding schedules?*

We agree that our experiments cannot discern between the effects of unknown oscillators responsible for feeding rhythms. Recent work suggests that feeding rhythms are associated with PER2 [19], but the fact they still exist in *Per1/2*-null mutants suggests there are other oscillators in play. Parasites could be responding to these food-driven oscillators as well as /or feeding directly. Our experiments were not designed to directly determine the precise rhythm that schedules parasites, but instead aimed to rule in or out the mammalian canonical clock, for which PER1 and PER2 have non-redundant roles [20]. We have modified the conclusion to clarify this.

Minor comments:

Methods:

The authors should add the JAX or other animal supplier catalog number for mouse strains so that experiments can be replicated if desired.

While the *Per1/2*-null mice used in this study are not available at JAX the *Per1*-null and *Per2*-null mutants they were derived from are available and the catalogue numbers have been added to (Lines 115-120).

There needs to be a better description of how they implemented daytime TRF? Did they add and remove food by hand with animals on the same cage, or instead did they change mice from a cage without food to a cage with food to avoid crumbs on the floor?

We added/removed food by hand, checking there were no pellets on cage floors. We did not change cages to avoid adding unnecessary handling stress to mice already under the stress of infection. We have clarified our TRF methodology (Lines 143-146). This method was also used for Prior et al 2018.

REFERENCES

1. Hirako I.C., Assis P.A., Hojo-Souza N.S., Reed G., Nakaya H., Golenbock D.T., Coimbra R.S., Gazzinelli R.T. 2018 Daily Rhythms of TNF α Expression and Food Intake Regulate Synchrony of Plasmodium Stages with the Host Circadian Cycle. *Cell Host Microbe* **23**(6), 796-808 e796. (doi:10.1016/j.chom.2018.04.016).
2. Prior K.F., van der Veen D.R., O'Donnell A.J., Cumnock K., Schneider D., Pain A., Subudhi A., Ramaprasad A., Rund S.S.C., Savill N.J., et al. 2018 Timing of host feeding drives rhythms in parasite replication. *PLoS Pathog* **14**(2).
3. Prior K.F., Rijo-Ferreira F., Assis P.A., Hirako I.C., Weaver D.R., Gazzinelli R.T., Reece S.E. 2020 Periodic Parasites and Daily Host Rhythms. *Cell Host Microbe* **27**(2), 176-187. (doi:<https://doi.org/10.1016/j.chom.2020.01.005>).
4. Maywood E., Chesham J., Smyllie N., Hastings M. 2014 The Tau mutation of casein kinase 1 ϵ sets the period of the mammalian pacemaker via regulation of Period1 or Period2 clock proteins. *J Biol Rhythms* **29**(2), 110-118.
5. Dudley C.A., Erbel-Sieler C., Estill S.J., Reick M., Franken P., Pitts S., McKnight S.L. 2003 Altered patterns of sleep and behavioral adaptability in NPAS2-deficient mice. *Science* **301**(5631), 379-383.
6. Pitts S., Perone E., Silver R. 2003 Food-entrained circadian rhythms are sustained in arrhythmic Clk/Clk mutant mice. *American Journal of Physiology-Regulatory, Integrative and Comparative Physiology* **285**(1), R57-R67.
7. Storch K.-F., Weitz C.J. 2009 Daily rhythms of food-anticipatory behavioral activity do not require the known circadian clock. *Proceedings of the National Academy of Sciences* **106**(16), 6808-6813.
8. Takasu N.N., Kurosawa G., Tokuda I.T., Mochizuki A., Todo T., Nakamura W. 2012 Circadian regulation of food-anticipatory activity in molecular clock-deficient mice. *PLoS One* **7**(11), e48892. (doi:10.1371/journal.pone.0048892).
9. Ogwan'g R.A., Mwangi J.K., Githure J., Were J., Roberts C.R., Martin S.K. 1993 Factors affecting exflagellation of in vitro-cultivated Plasmodium falciparum gametocytes. *The American journal of tropical medicine and hygiene* **49**(1), 25-29.
10. O'Donnell A.J., Mideo N., Reece S.E. 2014 Erratum to: disrupting rhythms in Plasmodium chabaudi: costs accrue quickly and independently of how infections are initiated. *Malar J* **13**(1), 503.
11. O'Donnell A.J., Mideo N., Reece S.E. 2013 Disrupting rhythms in Plasmodium chabaudi: costs accrue quickly and independently of how infections are initiated. *Malar J* **12**, 372. (doi:10.1186/1475-2875-12-372).
12. O'Donnell A.J., Schneider P., McWatters H.G., Reece S.E. 2011 Fitness costs of disrupting circadian rhythms in malaria parasites. *Proc Biol Sci* **278**(1717), 2429-2436. (doi:rsos.2010.2457/10.1098/rsos.2010.2457).

13. Kicska G.A., Ting L.M., Schramm V.L., Kim K. 2003 Effect of dietary p-aminobenzoic acid on murine *Plasmodium yoelii* infection. *J Infect Dis* **188**(11), 1776-1781. (doi:10.1086/379373).
14. Jacobs R.L. 1964 Role of p-aminobenzoic acid in *Plasmodium berghei* infection in the mouse. *Exp Parasitol* **15**(3), 213-225.
15. Taylor L.H., Walliker D., Read A.F. 1997 Mixed-genotype infections of the rodent malaria *Plasmodium chabaudi* are more infectious to mosquitoes than single-genotype infections. *Parasitology* **115**, 121-132.
16. Wale N., Sim D.G., Read A.F. 2017 A nutrient mediates intraspecific competition between rodent malaria parasites in vivo. *Proceedings of the Royal Society B: Biological Sciences* **284**(1859), 20171067.
17. Walliker D., Carter R., Morgan S. 1973 Genetic recombination in *Plasmodium berghei*. *Parasitology* **66**(2), 309-320. (doi:10.1017/s0031182000045248).
18. Hawking F. 1954 Milk, p-aminobenzoate, and malaria of rats and monkeys. *Br Med J* **1**(4859), 425.
19. Crosby P., Hamnett R., Putker M., Hoyle N.P., Reed M., Karam C.J., Maywood E.S., Stangherlin A., Chesham J.E., Hayter E.A. 2019 Insulin/IGF-1 drives PERIOD synthesis to entrain circadian rhythms with feeding time. *Cell* **177**(4), 896-909. e820.
20. Zheng B., Albrecht U., Kaasik K., Sage M., Lu W., Vaishnav S., Li Q., Sun Z.S., Eichele G., Bradley A., et al. 2001 Nonredundant roles of the mPer1 and mPer2 genes in the mammalian circadian clock. *Cell* **105**(5), 683-694.

Appendix B

Dear Professor Heesterbeek

Many thanks for sending our manuscript out to review and giving us the opportunity to revise it (RSPB-2020-0347) in light of the new reviewer's (reviewer 3) comments. We are grateful to the referees for their insight and constructive remarks.

We have revised our manuscript according to the reviewers' comments and suggestions, as outlined below. The word count of the revised manuscript is 7577.

Sincerely,

Aidan O'Donnell, Kimberley Prior and Sarah Reece

Referee: 3

- 1) **Summary: The IDC character is not properly validated with a sufficient number of time points. The key is to increase the number of time points for IDC. An analysis with JTK_CYCLE is needed to provide a nonparametric analysis of circadian rhythms because of the small number of time points (9) on the IDC character.**

The reviewer is correct that JTK_CYCLE is a non-parametric option for analysis of rhythmicity. However, the analysis used in our manuscript Circwave (harmonic regression) was designed for short and sparse data sets [1] and has been used in several other studies with the same 4 hour sampling interval and 24h sampling duration used in our experiment. These studies have been published in a range of disciplines, including (eg Nature Scientific Reports [2, 3], Plos One [4], Plos Genetics [5], Cell Host and Microbes [6], Neuroscience [7] and the Journal of Physiology [8]). The reliability of the F-test used in Circwave's harmonic regression method assumes that the model residuals are normally distributed [1]. We checked our model fits using residual plots and Shapiro-Wilk tests - they met this assumption as we stated in the Statistical Methods section (ESM). We agree that normality of residuals is hard to assess from small data sets and so, for completeness we have now added the results from analysis using JTK_CYCLE to the manuscript (Electronic Supplementary Material; Table S2). JTK_Cycle returns the same patterns as Circwave, but exacerbates the differences in amplitude estimate between the key treatment group comparisons (i.e. between parasites with dampened rhythms in *ad lib* fed *Per1/2*-null mutants compared to highly rhythmic parasites in the other 3 treatment groups). Thus, because Circwave returns more conservative results we focus on this approach for the main text.

With regard to the duration of the time series, if estimating period was our aim we would have followed parasites for 2 cycles. Instead, our hypotheses concerned synchrony and phase (timing) which are appropriate to assess from the data we collected.

- 2) **There are no molecular correlates (such as RNAs followed or proteins followed) to support the hypothesis of independent circadian rhythms in parasite.**

Why the reviewer makes this comment is unclear. If our aim was to demonstrate the presence of a circadian oscillator in malaria parasites, then this would be the usual approach. Although, we note that the earliest studies fulfilling features of circadian oscillators (entrainment, free running, and temperature compensation) all occurred before the genetics era began. The aim of our study stems

from evolutionary ecology, not mechanism - we are **not** testing for the presence of a circadian oscillator in the parasite, but probing which aspect of within-host rhythmicity drives periodicity in a parasite phenotype that underpins fitness. Directly quantifying the trait in question – via developmental stage assessed by morphology – is more relevant than using a molecular correlate as a proxy for the trait. We have explicitly stated this in the introduction to avoid further confusion.

- 3) A proper phase, period, and amplitude analysis is not given. For example, what is the period? A more thorough analysis of phase is needed. See, for example, Caranica et al. (2019) in *Yale Biology and Medicine* 92: 169.

Please see the response to comments '1' and '10' in which the reviewer expands on this query.

- 4) Even given that IDC is rhythmic under ad libitum feeding and the absence of a host clock, how can the authors exclude other host queues, such as body temperature?

We have dedicated a paragraph to discussion of body temperature as a time-of-day cue (lines 323-339).

We also elaborated on this in the response to reviewers in the previous revision, but perhaps this reviewer, being new to the paper, was not sent our response, which was:

"We agree that we do see temperature rhythms in our TRF mice that is close to a full inversion when compared to WT mice (ESM Figure S3b). This differs from Prior et al [9] in which no complete inversion was seen, perhaps due the differences in methodology e.g. mice strain (Prior used MF1 vs our C57/bl6), housing (Prior group housed animals while we housed mice individually) and differences in TRF timing (Prior provided food for 12 hours vs our 10 hours). However, it is well known that malaria parasites have an ability to respond to temperature changes (gametogenesis in transmission stages is partly triggered by temperature) [10]. Yet, despite these points, we doubt host temperature drives parasite rhythms because responding to absolute temperature in the mammalian host would only give parasites a 12hour resolution to their timekeeping (i.e. cold in the day vs warm at night), and thus we would expect to see lower stage amplitudes. Alternatively, parasites may respond to a sharp change in temperature (either using an increase or decrease in host temperature as a timing cue). This creates two scenarios, both of which we suspect are unlikely:

1) *Parasites are scheduled by a sharp increase in temperature: This is unlikely to be a reliable signal to parasites as there are several occasions in the 24hr period when temperature increases occur. For example, in ad lib fed WT mice, which are more analogous to natural conditions than the other groups, (ESM Figure S3b) temperature rises sharply in the early part of the light period and again early in the dark period. Thus, if parasites were scheduled solely by an increase in temperature, we would expect infections to stochastically follow the light or dark increases in temperature, resulting in 50% of infections in this treatment group having each, opposing, schedule. Instead we observe very repeatable parasite rhythms within in this group.*

2) *Parasites are scheduled by a sharp decrease in temperature: Again, this is unlikely to be a reliable signal. In ad lib WT mice, there are two drops in temperature spaced 3 hours apart during the light period, and in TRF WT mice there are two significant drops in temperature spaced 12 hours apart (early in the light period and early in the dark period; ESM Figure S3b).*

It is theoretically possible that parasites “ignore” a cue when they are at certain stages in the IDC, which enables them to respond to the “correct” cue at the right time, but this is not consistent with observations. For example, this strategy cannot explain the rescheduling of parasites that have been mismatched by 12 hours to the host’s rhythm, because the parasites wouldn’t “know” that the cue appearing in time with their IDC stage is incorrect and thus never reschedule ([9, 11-13] and this study in which parasites in TRF mice were initiated out of phase with the feeding rhythm).

We do recognize that our experiments were not specifically designed to dissect the parasites ability to align their rhythms with temperature changes in host, but our observations suggest Occam’s Razor is not supportive of temperature scheduling parasites. However, we have revised the discussion section to include the potential effects of temperature on parasite rhythms”

Given that the relevance of temperature rhythms to the IDC schedule is remote, we do not feel it appropriate to spend more than a paragraph discussing it and body temperature rhythms are presented in detail in Figures S1, S2 and S3b.

5) While the L/D cycle of the host was eliminated, what was done about entrainment to temperature? What record of cage temperatures was maintained?

We assume the reviewer is referring to entrainment by the host having a knock-on effect on the parasite via a rhythm that is different to internal body temperature (which is discussed in point 4).

First, all mice in all treatment groups and both experiments were housed in the same room. Thus, if environmental temperature rhythms were a driver of the IDC schedule, we would have observed the same IDC rhythms in all treatment groups within each experiment. Instead, we find rhythms become/remains significantly dampened in *ad lib* fed *Per1/2*-null mice compared to all other groups.

Second, our animal unit maintains strict climate control with room temperature at 21 ± 0.5 C. Temperature is considered a rather weak Zeitgeber of mammalian oscillators (including many rodent species); in constant light conditions, animals do not entrain consistently to temperature even when exposed to daily fluctuations of over 14C [14-17].

Third, given that *Per1/2*-null mice are unable to entrain to the strongest Zeitgeber – light - it’s remote that they entrain to temperature.

6) What are the RBC levels doing in the infected mice? Does this plasmodium generate recrudescence infections?

Please see the response to comment ‘11’ in which the reviewer elaborates on this point.

Specific Comments (L is line number)

7) L41. No molecular correlates are presented in the parasite.

We have responded to this comment above, please see comment ‘2’. Our question is not whether the parasite has an endogenous clock but about how within-host ecology shapes a periodic phenotype.

8) L136. What happens with different strains of plasmodium?

If the reviewer is asking whether there is genetic variation in how the IDC schedule responds to feeding rhythms, we agree that is a very interesting question. However, it is beyond the scope of this paper – an

equal sized study would have to be conducted for each strain of interest. The aim of this paper is to provide proof of principle, which is the first step before asking more in-depth questions about genetic variation.

However, we have added a comment in the discussion that we expect our findings to generalize across strains given the similarities in the IDC schedules observed in this study and in Hirako *et al* 2018 and Prior 2018 which used different strains of *Plasmodium chabaudi* [18, 19].

9) L168. Two complete cycles is needed at a minimum. Sample size is too small with 9 time points. In general there needs to be more time points and more cycles.

We agree with the reviewer that 2 cycles would be needed if we were to identify an endogenous parasite oscillator or measure period (response to comment '1'). However, our intention is to characterize IDC synchrony, timing, and fitness consequences of parasite rhythms and for this a snapshot is sufficient. We know from previous work (unpublished and [9, 20]) that the estimates we observe are consistent with estimates from a longer time series so we do not believe there is a lack of power for detecting rhythmicity. Further, sampling mice every 4 hours for more than two cycles would influence their behavior and increase disease severity, potentially biasing our virulent measures.

10) L191. There is a need to report phase, period, and amplitude with standard errors. There is a need to report percentage variation explained by each COSINOR analysis. If $A\cos(\omega t + \varphi) + \text{error}$ is being fitted, say so. Also what is the percentage variation explained by this simple model when the nonlinear regression on time is done?

Model fits determining periodicity were obtained using Circwave. Circwave uses a forward linear harmonic regression. A sinusoidal wave is fitted to the data and is described by the formula.

$$f(t) = a + \sum_{i=1}^{\infty} \left[p_i \sin i 2\pi \frac{t}{\tau} + q_i \cos i 2\pi \frac{t}{\tau} \right]$$

Where

- $f(t)$ = the calculated function value at time point t
- a = mean;
- i = indicating the harmonic wave,
- p_i = the sine coefficient harmonic
- q_i = coefficient of the harmonic
- t = timepoint value (modulo τ);

The wave is then tested against a fitted horizontal line of the average. We reported our results following the convention of other papers using Circwave but have now added full details for the model fits (including F ratio, P value and R²) in supplementary material (Table S1). R² regression fits were strong (between 0.68 and 0.98).

The parameters of phase and amplitude the reviewer requests are already in the manuscript. Circadian amplitudes were calculated from the regression model fits ($\sqrt{a^2+b^2}$) and reported with standard errors (see lines 199-208 and 224-232). Circadian phases were calculated from harmonic regression model fits ($\arctan(a + b)$) and reported as circular means with circular standard deviations. Period was set at a fixed tau of 24 because it is well known that period measures are unreliable without data from

multiple cycles – and the predictions under test were not about period. However, we have now included the Harmonic regression model results in the supplementary material (Table S1).

11) L207. What is RBC doing over time in the mice? Does this plasmodium cause a recrudescence infection, and what is its period? Some recrudescence infections in primate species and with particular plasmodium species can cause recrudescence infections with a period of 24 h.

RBC's show no evidence of daily rhythmicity in the sampling window, rather there is a steady linear decline across the time series. It is true that healthy mice have some circadian variation in RBC density but this is swamped by the destructive effects of malaria parasite replication.

We are not clear on how the reviewer is defining recrudescence. In malariology, "recrudescence" refers to small peaks in parasite replication that occur once the acute infection is controlled by the host. *P. chabaudi* recrudescence occurs after approx. 21 days and patency occurs irregularly for bouts of a few days in each recrudescence [21, 22]. However, we are not aware of any evidence that IDC duration varies in acute phase infection and recrudescence, if this is what the reviewer is wondering? Alternatively, perhaps the reviewer is referring to *P. knowlesi*, a zoonotic species of malaria that infects primates and has an IDC duration of 24 hours [23].

12) L208. With only 9 time points a nonparametric method is needed, such as JTK_CYCLE.

Please see the response to comment '1' in which the reviewer elaborates on this comment

13) L255. To reach the conclusion that IDC has a clock additional experiments are needed. It must be shown that IDC not only has a period of 24 h (which is not reported in this paper), but that IDC is subject to Zeitgeber entrainment and temperature compensation. What about these other properties? Does the IDC respond to a different light schedule or food schedule for the host?

We are confused as to why the reviewer feels the aim of the paper is to test whether the IDC is driven by a circadian oscillator. When we give talks on our lab's work, most mechanism-researching chronobiologists focus on whether the parasite has a clock or not. We have reasons to think not, but moreover, we are evolutionary ecologists and are more motivated by explaining host-parasite interactions than uncovering genes and molecular mechanisms. In the introduction, we state our aim as "we use the rodent malaria parasite *P. chabaudi* to test whether the host feeding-associated rhythm that drives the IDC schedule relies upon - or is independent of - the Transcription-Translation-Feedback-Loops that form a major part of the host's circadian clock mechanism". It is unclear how this can be interpreted as asking if the parasite has an endogenous oscillator. We discuss that the parasite is likely to be, at least in part, in control of the IDC schedule but this does not mean it must use a clock. Far more simple cue-response systems exist that allow organisms to respond to time-of-day changes in the environment that do not have the properties of clocks. Finally, previous work published in Prior *et al* 2018 and Hirako *et al* 2018 demonstrate that the light:dark schedule does not set the timing of the IDC and that altering the timing of food provision alters the timing of the IDC accordingly [9, 18]. These findings were already explained in the manuscript.

Referee: 2

In this revised version of the manuscript by Odonnell et al., the authors made significant changes that in our view have greatly improved the scientific accuracy and support of their conclusions. Except for some minor points highlighted below, all other concerns have been addressed.

1) Line 295: “food anticipatory activity (FAA; which requires an endogenous oscillator)”

This needs to be rephrased to clarify to which oscillator they are referring. SCN-ablated animals still exhibit normal FAA, indicating that the food-entrainable oscillator is located outside of the SCN (Mistlberger, 1994). It does not seem to rely on the canonical molecular circadian timekeeping mechanism. However, without clarifying which oscillator is meant, the authors contradict themselves in the same sentence: “food anticipatory activity (FAA; which requires an endogenous oscillator) can be observed in clock-disrupted mice [40]”

We have clarified the discussion as per the reviewer’s suggestion.

2) The authors refer to Food Anticipatory activity in arrhythmic mutant mice upon restricted feeding, as an example. It would be good to clarify that 12h time restriction has also been shown to rescue rhythmicity in arrhythmic mice, e.g. gene expression rhythmicity upon time restricted feeding in mutant mice (doi.org/10.1073/pnas.1515308112). The authors should read the circadian literature and include additional relevant citations. There are studies from the Schibler lab and Panda lab, among others, that have shown this. Thus, it is not surprising that feeding rhythms can entrain the parasite population since they are entraining host rhythms, and host rhythms (as a whole without knowing exactly which cues) have been shown to entrain parasite population rhythms. This is now acknowledged by the authors, but the manuscript would benefit with them clarifying that this exact protocol was expected to show the observed results.

These references were removed due to length constraints but we do agree with the reviewer and so have found space to expand the citations in the discussion.

3) We appreciate that the authors attempted to verify that their assessment of activity was a good proxy for food intake. However, we feel this still requires further work:

a) There is no axis information for Fig. S4

Figure S4 has been replaced by a new figure which now includes a quantitative representation of the intensity of activity and food bouts, rather than binary measures.

b) Mice are group housed, so is this information from a single mouse? More detailed information is needed about what is represented in the figure.

All measures were summed across 5 individuals group housed in the cage. We have clarified this in the figure legend.

c) We do not believe this is the best analysis/representation because there is no y axis. It seems that a small, 1 min movement would count the same as a 15 min active bout, which leads to the impression that mice are hyperactive. Perhaps a heatmap of some type, or simply including a clearly labelled y axis would avoid confusion.

Please see response to comment 3a.

4) Please add details in each figure legend about the day of infection measured, and how many mice are represented in each figure for easier understanding without having to refer to the manuscript.

We have amended figure legends to include Day PI and mouse numbers as suggested.

References:

1. Cornelissen G. 2014 Cosinor-based rhythmometry. *Theor Biol Med Model* **11**(1), 16.
2. Bertolini E., Kistenpfennig C., Menegazzi P., Keller A., Koukidou M., Helfrich-Förster C. 2018 The characterization of the circadian clock in the olive fly *Bactrocera oleae* (Diptera: Tephritidae) reveals a *Drosophila*-like organization. *Scientific reports* **8**(1), 816. (doi:10.1038/s41598-018-19255-8).
3. Zhou P., Ross R.A., Pywell C.M., Liangpunsakul S., Duffield G.E. 2014 Disturbances in the murine hepatic circadian clock in alcohol-induced hepatic steatosis. *Scientific reports* **4**, 3725.
4. da Silva Santos E.A., Marques T.E.B.S., de Carvalho Matos H., Leite J.P., Garcia-Cairasco N., Paçó-Larson M.L., Gitaí D.L.G. 2015 Diurnal variation has effect on differential gene expression analysis in the hippocampus of the pilocarpine-induced model of mesial temporal lobe epilepsy. *PLoS One* **10**(10).
5. Landgraf D., Wang L.L., Diemer T., Welsh D.K. 2016 NPAS2 compensates for loss of CLOCK in peripheral circadian oscillators. *PLoS Genet* **12**(2).
6. Leone V., Gibbons S.M., Martinez K., Hutchison A.L., Huang E.Y., Cham C.M., Pierre J.F., Heneghan A.F., Nadimpalli A., Hubert N. 2015 Effects of diurnal variation of gut microbes and high-fat feeding on host circadian clock function and metabolism. *Cell Host Microbe* **17**(5), 681-689.
7. Ventzke K., Oster H., Jöhren O. 2019 Diurnal Regulation of the Orexin/Hypocretin System in Mice. *Neuroscience* **421**, 59-68.
8. Gatford K.L., Kennaway D.J., Liu H., Kleemann D.O., Kuchel T.R., Varcoe T.J. 2019 Simulated shift work disrupts maternal circadian rhythms and metabolism, and increases gestation length in sheep. *The Journal of physiology* **597**(7), 1889-1904.
9. Prior K.F., van der Veen D.R., O'Donnell A.J., Cumnock K., Schneider D., Pain A., Subudhi A., Ramaprasad A., Rund S.S.C., Savill N.J., et al. 2018 Timing of host feeding drives rhythms in parasite replication. *PLoS Pathog* **14**(2).
10. Ogwan'g R.A., Mwangi J.K., Githure J., Were J., Roberts C.R., Martin S.K. 1993 Factors affecting exflagellation of in vitro-cultivated *Plasmodium falciparum* gametocytes. *The American journal of tropical medicine and hygiene* **49**(1), 25-29.
11. O'Donnell A.J., Mideo N., Reece S.E. 2014 Erratum to: disrupting rhythms in *Plasmodium chabaudi*: costs accrue quickly and independently of how infections are initiated. *Malar J* **13**(1), 503.
12. O'Donnell A.J., Mideo N., Reece S.E. 2013 Disrupting rhythms in *Plasmodium chabaudi*: costs accrue quickly and independently of how infections are initiated. *Malar J* **12**, 372. (doi:10.1186/1475-2875-12-372).
13. O'Donnell A.J., Schneider P., McWatters H.G., Reece S.E. 2011 Fitness costs of disrupting circadian rhythms in malaria parasites. *Proc Biol Sci* **278**(1717), 2429-2436. (doi:rsos.2010.2457 [pii] 10.1098/rsos.2010.2457).
14. Francis A.J., Coleman G.J. 1997 Phase response curves to ambient temperature pulses in rats. *Physiol Behav* **62**(6), 1211-1217.

15. Pohl H.H. 1998 Temperature cycles as zeitgeber for the circadian clock of two burrowing rodents, the normothermic antelope ground squirrel and the heterothermic Syrian hamster. *Biol Rhythm Res* **29**(3), 311-325.
16. Refinetti R. 2010 Entrainment of circadian rhythm by ambient temperature cycles in mice. *J Biol Rhythms* **25**(4), 247-256.
17. Refinetti R. 2015 Comparison of light, food, and temperature as environmental synchronizers of the circadian rhythm of activity in mice. *The Journal of Physiological Sciences* **65**(4), 359-366.
18. Hirako I.C., Assis P.A., Hojo-Souza N.S., Reed G., Nakaya H., Golenbock D.T., Coimbra R.S., Gazzinelli R.T. 2018 Daily Rhythms of TNFalpha Expression and Food Intake Regulate Synchrony of Plasmodium Stages with the Host Circadian Cycle. *Cell Host Microbe* **23**(6), 796-808 e796. (doi:10.1016/j.chom.2018.04.016).
19. Prior K.F. 2018 The evolutionary ecology of circadian rhythms in malaria parasites, University of Edinburgh.
20. Prior K.F., Rijo-Ferreira F., Assis P.A., Hirako I.C., Weaver D.R., Gazzinelli R.T., Reece S.E. 2020 Periodic Parasites and Daily Host Rhythms. *Cell Host Microbe* **27**(2), 176-187. (doi:<https://doi.org/10.1016/j.chom.2020.01.005>).
21. Spence P.J., Jarra W., Lévy P., Reid A.J., Chappell L., Brugat T., Sanders M., Berriman M., Langhorne J. 2013 Vector transmission regulates immune control of Plasmodium virulence. *Nature* **498**(7453), 228-231.
22. Stephens R., Culleton R.L., Lamb T.J. 2012 The contribution of Plasmodium chabaudi to our understanding of malaria. *Trends Parasitol* **28**(2), 73-82.
23. Coatney G.R. 1971 *The primate malarial*, US National Institute of Allergy and Infectious Diseases.

Appendix C

Review of “Host circadian clocks do not set the schedule for the within host replication of malaria parasites”

Reviewer 3 has focused on the responses by the authors to several points raised by reviewer 3. Reviewer 3 in turn below responds to these comments.

- 1) **Summary: The IDC character is not properly validated with a sufficient number of time points. The key is to increase the number of time points for IDC. An analysis with JTK_CYCLE is needed to provide a nonparametric analysis of circadian rhythms because of the small number of time points (9) on the IDC character.**

The standard size of an experiment is 48 h. Examples include any of the papers by Jay Dunlap and colleagues. See, for example, Crosthwaite et al. (1995) in *Cell* 81: 1004 or in *Science* 276: 763 (1997). The problem with the shorter period for the experiment is the inability to examine period variation. For example, in the JTK_CYCLE and harmonic regression analysis the authors had to fix the period at 24 h. The period is relevant in distinguishing a feeding rhythm from a circadian rhythm in the parasite to explain the observations on IDC. Also the phase is the noisiest part of the characterization usually and is part of their paper’s point about synchronization. Having more time points will strengthen the synchronization argument.

Harmonic regression is actually meant for much longer time series. See, for example, F. J. Anscombe’s *Computing in Statistical Science with APL*. In this work in Ch. 10 harmonic regression is used for 100s if not 1000s of observations. This is the appropriate context for testing the normality assumption underlying harmonic regression using Wilks-Shapiro.

- 2) **L136. What happens with different strains of plasmodium?**

The reason that this question is raised is simply to know how general the findings are in the paper.

- 3) **L41 There are no molecular correlates (such as RNAs followed or proteins followed) to support the hypothesis of independent circadian rhythms in parasite.**

The authors put forward the hypothesis that rhythmic feeding is driving the oscillations in the parasites. An alternative equally plausible hypothesis is that the parasite has circadian rhythms to anticipate rhythmic availability of food. That is why it is relevant to know something about molecular correlates in the parasite.

- 4) **L168. Two complete cycles is needed at a minimum. Sample size is too small with 9 time points. In general there needs to be more time points and more cycles.**

The problem is that a circadian oscillator in the parasite could explain the authors’ data as an alternative hypothesis; therefore, the period observed is highly relevant to distinguishing a feeding rhythm hypothesis versus a circadian rhythm in the parasite hypothesis.

- 5) L207. What is RBC doing over time in the mice? Does this plasmodium cause a recrudescence infection, and what is its period? Some recrudescence infections in primate species and with particular plasmodium species can cause recrudescence infections with a period of 24 h.

The *P. knowlesi* case is what reviewer 3 had in mind. The reviewer wanted to be sure that the author's choice of Plasmodium in this host did not fall in the same category as *P. knowlesi*. That the authors' choice of strain/host combination does not fall in this category should probably be mentioned in the text.

- 6) L255. To reach the conclusion that IDC has a clock additional experiments are needed. It must be shown that IDC not only has a period of 24 h (which is not reported in this paper), but that IDC is subject to Zeitgeber entrainment and temperature compensation. What about these other properties? Does the IDC respond to a different light schedule or food schedule for the host?

How can the authors talk about synchronized feeding driving the oscillations seen in the parasite without entertaining the alternative hypothesis that the parasite has a circadian rhythm anticipating such feeding schedules? It is in this context that having more time points is relevant. Being able to estimate period of IDC then becomes relevant for distinguishing these two hypotheses.

Material below is relevant to the revised manuscript.

Summary: Reviewer 3 does not see how circadian rhythms in the parasite can be set aside based on the experiments here. Either rhythmic feeding schedule or circadian rhythms in the parasite anticipating rhythmic feeding schedules could explain the results. There is also a problem with the chi-squared statistics reported here that needs to be resolved. Temperature entrainment experiments need to be considered to rule out an alternative cue to feeding schedule.

Specific Comments (L is line number)

L25. The authors cannot rule out the role of a possible parasite circadian rhythm.

L38. It is interesting that the authors discuss circadian rhythms in the pathogen *Botrytis cinerea*. Why doesn't this example support examining the same hypothesis here?

L69. How do the parasites schedule their IDC? It could be using a parasite clock or an environmental cue, such as feeding.

L90. It is hard to test the role of environmental cues without entertaining the effects of circadian rhythm anticipating these cues.

L198. The chi-squared statistic makes no sense here with 15 df. It should not have a $P < .001$. The mean of the chi-squared statistic under the null is around 15. So, a chi-squared of .43 is well below the cutoff of 25.00 at the 0.05 level.

L211. I have similar problems with the chi-squared statistics reported here. They are very large in magnitude with respect to the df and should be significant at any reasonable alpha level. I have similar problems in the supplements.

L217 Same problem here with too large a chi-squared statistic reported to be considered insignificant.

L222 The chi-squared statistic is small, but it is reported as significant. Don't understand what is going on here.

L257. It would be nice to know that the results do generalize across strains.

L281. Normally in mammals there is glucose compensation, although one might argue that this phenomenon may not make sense in a microbe. I think the authors make the key point that even if there is glucose compensation normally, the presence of the parasite may shut down this mechanism. So, glucose signaling could be involved as an environmental signal.

L320. It would be interesting to test whether or not temperature could be used as an entrainment signal.

Appendix D

Dear Professor Heesterbeek

We are grateful for the opportunity to revise the manuscript (RSPB-2020-0347.R1) and thank the reviewers for their continued attention to it. Both reviewer 1 and 2 are happy with the manuscript as it stands and we have made extensive changes in response to reviewer 3's comments.

We have never had a paper cause so much confusion with a reviewer before. We think we have finally identified a sentence in the abstract and the use of terms from evolutionary ecology that can be interpreted by readers from other disciplines in ways that are not our intention. We think this explains the disconnect between our intention and Reviewer 3's interpretation. Thus, we have altered the abstract, intro, and discussion to clarify our aims, by using terminology that should be clear across disciplines.

The central issue that the remaining reviewer (#3) has, stems from their belief that we dismiss the notion that malaria parasites might possess an endogenous clock. Identifying clocks and how they function are core activities in chronobiology and so we suspect this may be reviewer 3's perspective. Put another way, it seems the reviewer assumes our paper is about testing for the presence of a parasite clock. However, we have been careful to avoid speculation about the mechanism underpinning rhythmicity in the parasite because our data do not directly speak to this issue. Our experiments are designed to determine which kind of host rhythm the parasite's rhythm is aligned with, not how this alignment is achieved. We think this misunderstanding also explains why the reviewer is critical of our approach – we agree our approach is not the right way to test for a clock, but as that is not our aim, our approach is not flawed. We have again explained this in the response letter and conveyed our aims more clearly in the revised manuscript.

We are aware that another group are preparing a paper with very similar experiments (just using a different kind of clock mutant mouse host for malaria parasites) and they find the same results. Our paper was first submitted to Proc Roy Soc B back in September 2019 and has experienced long delays at each stage whilst out to review. Thus, we would be grateful if the Editor/ Associate Editor are able to make a decision without the time costs involved in soliciting further reviews.

Many thanks,

Aidan O'Donnell, Kim Prior, & Sarah Reece

Reviewer 3:

The reviewer's central concern is that we conclude malaria parasites do not have a circadian clock, and that this conclusion is not clearly supported by our data.

Our experiments are designed to determine which kind of host rhythm the parasite's rhythm is aligned with, not how this alignment is achieved (i.e. whether by a parasite clock or any other scheduling method). This also explains why the reviewer is critical of our approach – we agree our approach is not the right way to test for a clock, but, as we explained in our previous response, that is not our aim. However, we now realise that a sentence in the abstract and some of the terms we used could be interpreted in different ways. For example, when we have used the term "driven" we do not mean to suggest that the host is imposing a rhythm upon intrinsically arrhythmic parasites by killing out-of-synch IDC stages.

To remove this confusion, we have substantially re-written the abstract, introduction, and discussion to convey our aims and conclusions more clearly, including a change in terminology. This includes specifying throughout whether we are referring to a parasite or host rhythm when use the terms feeding-related rhythms and clock-driven rhythms. The re-write also emphasizes that our approach is designed to probe the nature of the interaction between host and parasite rhythms, not the type of mechanism(s) that underpins this interaction. E.g. we now write "*We test if rhythmicity in parasite replication is coordinated with the host's feeding-related rhythms and/or rhythms driven by the host's canonical circadian clock*". And "*We conclude that malaria parasite replication is coordinated to rhythmic host processes that are independent of the core-clock proteins PERIOD 1 and 2; mostly likely a periodic nutrient made available when the host digests food.*" However, seeing as the reviewer desires discussion of whether the parasites have a clock or not, we have edited the conclusions section to address to what extent our data are consistent with parasites keeping time

themselves, via a clock or simpler response to time-of-day. *“The speed with which the IDC schedule changes, its precision, and the modest initial loss of parasite number involved in rescheduling [12] along with parasite control of IDC duration [24], suggest the parasite is actively aligning certain developmental stages with host feeding rhythms to take advantage of periodicity in a resource(s) it must acquire from the host’s processing of its food. Recent studies suggest that IDC rhythms display a hallmark of a circadian clock [9,10] but other criteria (temperature compensation, entrainment) are yet to be met. Whatever the parasites’ method of time-keeping, our data suggest it uses a signal stemming from the host’s processing of food as a Zeitgeber or timing cue.”*

Responses to other comments:

- 1) *“The standard size of an experiment is 48 h. Examples include any of the papers by Jay Dunlap and colleagues. See, for example, Crosthwaite et al. (1995) in Cell 81: 1004 or in Science 276: 763 (1997). The problem with the shorter period for the experiment is the inability to examine period variation. For example, in the JTK_CYCLE and harmonic regression analysis the authors had to fix the period at 24 h. The period is relevant in distinguishing a feeding rhythm from a circadian rhythm in the parasite to explain the observations on IDC. Also the phase is the noisiest part of the characterization usually and is part of their paper’s point about synchronization. Having more time points will strengthen the synchronization argument. Harmonic regression is actually meant for much longer time series. See, for example, F. J. Anscombe’s Computing in Statistical Science with APL. In this work in Ch. 10 harmonic regression is used for 100s if not 1000s of observations. This is the appropriate context for testing the normality assumption underlying harmonic regression using Wilks-Shapiro*

The issue here is that *“The period is relevant in distinguishing a feeding rhythm from a circadian rhythm in the parasite”*. As explained above, we are not attempting to distinguish between the parasite using a ‘circadian clock (or other kind of time-keeping mechanism) to couple its IDC to the timing of host feeding’ versus ‘a nutrient from the host’s food enforcing a rhythm upon the IDC via mistimed parasites starving to death’. Our data, across our treatment groups and in both experiments, reveal that the IDC rhythm is aligned – by an unknown mechanism – to a rhythm stemming from the timing of host feeding alone, and not to a host rhythm that depends on the host’s canonical circadian clock. Put another way, if the parasite has a clock (or other time-keeping mechanism) then it is using a signal provided by the host’s processing of food as a Zeitgeber (time-cue). The conclusions section now states: *“Whatever the parasites’ method of time-keeping, our data suggest it uses a signal stemming from the host’s processing of food as a Zeitgeber or timing cue”*. In contrast, if we were probing for the presence of a clock then yes, we would examine a much longer time-series because estimating period would be necessary (which would tradeoff against the frequency of samples that could be taken). We also note that other studies provide ample evidence that the period of the IDC matches that of the host’s rhythm [1-5]. Furthermore, revealing that the period of the IDC matches that of host rhythms (i.e. is approx. 24h) would not help differentiate which host rhythm the IDC is aligned with because the feeding rhythms we impose have a period of 24 hours and WT hosts have circadian rhythms with a period of 24 hours. We designed our study to decouple the timing (phase) of feeding and TTFL-clock rhythms in the host, and thus, use phase and synchrony of the IDC to test which host rhythms it aligns with.

In response to *“Also the phase is the noisiest part of the characterization usually and is part of their paper’s point about synchronization”*, we urge the reviewer to consult the results section again. We do not use phase as a measure of synchrony. We report phase (timing of the parasites rhythm) in relation to the phase of the host’s feeding rhythms and we report amplitude as a measure of synchrony of the parasite rhythm. Moreover, that the phase of the parasite rhythm aligns with host feeding rhythms across our treatment groups and in both experiments, despite its “noise” makes our results conservative.

- 2) *The authors put forward the hypothesis that rhythmic feeding is driving the oscillations in the parasites. An alternative equally plausible hypothesis is that the parasite has circadian rhythms to anticipate rhythmic availability of food. That is why it is relevant to know something about molecular correlates in the parasite.*

As explained above and in the general comments, our aim is not to speculate on **how** the parasite IDC is aligned with host feeding rhythms (i.e. the underlying parasite mechanisms). It seems that the reviewer has interpreted our conclusion to be that the parasite is intrinsically arrhythmic and that the host is enforcing a rhythm on the IDC, by for example, starvation of mistimed IDC stages. This might be the reality – although we doubt it is because most evidence, including from our lab [4-8] suggests the parasite is at least in part responsible for scheduling its own IDC. Instead, we

propose that if the parasite has a clock (or other time-keeping mechanism) then it is using a signal provided by the host's processing of food as a Zeitgeber (time-cue). We have altered the conclusion section to make it clear that the parasite probably has some way to tell the time but this cannot be assumed based on our data or that of recent papers. *"The speed with which the IDC schedule changes, its precision, and the modest initial loss of parasite numbers involved in rescheduling [12] along with parasite control of IDC duration [24], suggest the parasite is actively aligning certain developmental stages with host feeding rhythms to take advantage of periodicity in a resource(s) it must acquire from the host's processing of its food. Recent studies suggest that IDC rhythms display a hallmark of a circadian clock [9,10] but other criteria (temperature compensation, entrainment) are yet to be met."*

More generally, it is not clear what the reviewer means by "molecular correlates" (*"That is why it is relevant to know something about molecular correlates in the parasite."*). Demonstrating rhythmicity in expression, or at the protein level, of clock genes would – of course - indicate a clock, but this is circular because it first requires identifying clock genes. Alternatively, if the reviewer means the common approach used in chronobiology to generate a reporter cell line (e.g. using a luciferase tag) as a readout of a clock, then this would be a readout of a potential clock, but likewise, the IDC phenotype is also an output of a potential clock. Given that our aim is to explain the IDC phenotype, it is not clear what would be gained from studying this trait indirectly. Furthermore, recent papers [4, 5, 7] provide plenty of evidence that gene expression of the parasites are rhythmic (with a period that matches the host's period) – but no clock genes have been identified from these hits and the overwhelming majority are rhythmic simply due to the temporal cascade of developmental changes during the IDC.

3) The reason that this question is raised is simply to know how general the findings are in the paper.

We believe this comment refers to the first review in which the reviewer queried whether our findings are generalizable across strains. We addressed this in the revision and added the sentence *"We expect our findings to generalise across strains given the similarities in the IDC schedules observed in this study and in Hirako et al 2018 [18] and Prior 2018 [31] which used different strains of P. chabaudi."* to the paper. It is not clear to us whether the reviewer is justifying why they queried this in the first review or if they are not satisfied by our addition to the paper.

4) The problem is that a circadian oscillator in the parasite could explain the authors' data as an alternative hypothesis; therefore, the period observed is highly relevant to distinguishing a feeding rhythm hypothesis versus a circadian rhythm in the parasite hypothesis.

Please refer to the general comment and responses to points (1) and (2).

5) The *P. knowlesi* case is what reviewer 3 had in mind. The reviewer wanted to be sure that the author's choice of Plasmodium in this host did not fall in the same category as *P. knowlesi*. That the authors' choice of strain/host combination does not fall in this category should probably be mentioned in the text.

We have now included reference to 3 recent papers [4, 5, 7] that reveal the same principles underpin rhythmicity of the IDC in human and rodent parasites, including the species used in our paper.

6) How can the authors talk about synchronized feeding driving the oscillations seen in the parasite without entertaining the alternative hypothesis that the parasite has a circadian rhythm anticipating such feeding schedules? It is in this context that having more time points is relevant. Being able to estimate period of IDC then becomes relevant for distinguishing these two hypotheses.

Please refer to the general comment and responses to points (1) and (2).

Specific Comments (L is line number)

L25. The authors cannot rule out the role of a possible parasite circadian rhythm. L38. It is interesting that the authors discuss circadian rhythms in the pathogen *Botrytis cinerea*. Why doesn't this example support examining the same hypothesis here? L69. How do the parasites schedule their IDC? It could be using a parasite clock or an environmental cue, such as feeding. L90. It is hard to test the role of environmental cues without entertaining the effects of circadian rhythm anticipating these cues.

We have combined these comments as they all stem from the same idea that parasites may have an intrinsic oscillator. We agree that this is possible but as explained previously and above, our intention is not to uncover the mechanism that is driving the parasite rhythm. Rather our paper reveals that host feeding is a cue for the IDC schedule, either in the form of a Zeitgeber for an intrinsic parasite clock or as a time-of-day cue for a simpler “just-in-time” reactionary mechanism.

L198. The chi-squared statistic makes no sense here with 15 df. It should not have a $P < .001$. The mean of the chi-squared statistic under the null is around 15. So, a chi-squared of .43 is well below the cutoff of 25.00 at the 0.05 level. L211. I have similar problems with the chi-squared statistics reported here. They are very large in magnitude with respect to the df and should be significant at any reasonable alpha level. I have similar problems in the supplements. L217 Same problem here with too large a chi-squared statistic reported to be considered insignificant. L222 The chi-squared statistic is small, but it is reported as significant. Don't understand what is going on here.

Previously, we reported the change in deviance (the difference in $2 \times (\log \text{likelihood})$ between a saturated model and the actual model) as these values are synonymous with a chi-square value when comparing generalised linear models [9]. We apologise that we were not explicit in pointing out that we were reporting deviance, and we agree that this can make it difficult to interpret the goodness of fits for the models (as the deviance scales with the data). We have re-analysed the models using F-tests and have made interpretation of goodness of fits more intuitive by reporting F-values throughout the manuscript (with the exception of the ESM in which Chi-square values for linear mixed effect models with mouse ID as a random effect were now used).

L257. It would be nice to know that the results do generalize across strains.

Please refer to the response to point (3).

L281. Normally in mammals there is glucose compensation, although one might argue that this phenomenon may not make sense in a microbe. I think the authors make the key point that even if there is glucose compensation normally, the presence of the parasite may shut down this mechanism. So, glucose signaling could be involved as an environmental signal.

The idea that glucose could provide environmental time of day information to the parasite is proposed by Prior et al [3] and Hirako et al [1]. These studies investigated well-developed infections and the glucose rhythm observed relies on highly active immune responses. Thus, a strong glucose rhythm is unlikely to operate at the start of infections, before immune responses are well developed. Given that the IDC can become rescheduled in the first few days of infections, it is unlikely that glucose is responsible – either by enforcing an IDC rhythm and/or by providing time-of-day information to the parasite [6].

L320. It would be interesting to test whether or not temperature could be used as an entrainment signal.

We provided an extensive rebuttal of this possibility in the first revision and included new figures and a dedicated section in the discussion section of the manuscript. Our reasoning for ruling our temperature is copied from the previous rebuttal below, but it is not clear whether the reviewer disagrees (and if so, with which aspects).

“We have dedicated a paragraph to discussion of body temperature as a Zeitgeber/time-of-day cue (lines 301-316).

“We agree that we do see temperature rhythms in our TRF mice that is close to a full inversion when compared to WT mice (ESM Figure S3b). This differs from Prior et al [3] in which no complete inversion was seen, perhaps due the differences in methodology e.g. mice strain (Prior used MF1 vs our C57/bl6), housing (Prior group housed animals while we housed mice individually) and differences in TRF timing (Prior provided food for 12 hours vs our 10 hours). However, it is well known that malaria parasites have an ability to respond to temperature changes (gametogenesis in transmission stages is partly triggered by temperature) [10]. Yet, despite these points, we doubt host temperature drives parasite rhythms because responding to absolute temperature in the mammalian host would only give parasites a 12hour resolution to their timekeeping (i.e. cold in the day vs warm at night), and thus we would expect to see lower stage amplitudes. Alternatively, parasites may respond to a sharp change in temperature (either using an increase or decrease in host temperature as a timing cue). This creates two scenarios, both of which we suspect are unlikely:

1) *Parasites are scheduled by a sharp increase in temperature: This is unlikely to be a reliable signal to parasites as there are several occasions in the 24hr period when temperature increases occur. For example, in ad lib fed WT mice, which are more analogous to natural conditions than the other groups, (ESM Figure S3b) temperature rises sharply in the early part of the light period and again early in the dark period. Thus, if parasites were scheduled solely by an increase in temperature, we would expect infections to stochastically follow the light or dark increases in temperature, resulting in 50% of infections in this treatment group having each, opposing, schedule. Instead we observe very repeatable parasite rhythms within in this group.*

2) *Parasites are scheduled by a sharp decrease in temperature: Again, this is unlikely to be a reliable signal. In ad lib WT mice, there are two drops in temperature spaced 3 hours apart during the light period, and in TRF WT mice there are two significant drops in temperature spaced 12 hours apart (early in the light period and early in the dark period; ESM Figure S3b).*

It is theoretically possible that parasites “ignore” a cue when they are at certain stages in the IDC, which enables them to respond to the “correct” cue at the right time, but this is not consistent with observations. For example, this strategy cannot explain the rescheduling of parasites that have been mismatched by 12 hours to the host’s rhythm, because the parasites wouldn’t “know” that the cue appearing in time with their IDC stage is incorrect and thus never reschedule ([3, 11-13] and this study in which parasites in TRF mice were initiated out of phase with the feeding rhythm).

Given that the relevance of temperature rhythms to the IDC schedule is remote, we do not feel it appropriate to spend more than a paragraph discussing it and body temperature rhythms are presented in detail in Figures S1, S2 and S3b.

References:

1. Hirako I.C., Assis P.A., Hojo-Souza N.S., Reed G., Nakaya H., Golenbock D.T., Coimbra R.S., Gazzinelli R.T. 2018 Daily Rhythms of TNF α Expression and Food Intake Regulate Synchrony of Plasmodium Stages with the Host Circadian Cycle. *Cell Host Microbe* **23**(6), 796-808 e796. (doi:10.1016/j.chom.2018.04.016).
2. Prior K.F., Rijo-Ferreira F., Assis P.A., Hirako I.C., Weaver D.R., Gazzinelli R.T., Reece S.E. 2020 Periodic Parasites and Daily Host Rhythms. *Cell Host Microbe* **27**(2), 176-187. (doi:10.1016/j.chom.2020.01.005).
3. Prior K.F., van der Veen D.R., O'Donnell A.J., Cumnock K., Schneider D., Pain A., Subudhi A., Ramaprasad A., Rund S.S.C., Savill N.J., et al. 2018 Timing of host feeding drives rhythms in parasite replication. *PLoS Pathog* **14**(2).
4. Rijo-Ferreira F., Acosta-Rodriguez V.A., Abel J.H., Kornblum I., Bento I., Kilaru G., Klerman E.B., Mota M.M., Takahashi J.S. 2020 The malaria parasite has an intrinsic clock. *Science* **368**(6492), 746-753.
5. Subudhi A.K., O'Donnell A.J., Ramaprasad A., Abkhallo H.M., Kaushik A., Ansari H.R., Abdel-Haleem A.M., Ben Rached F., Kaneko O., Culleton R., et al. 2020 Malaria parasites regulate intra-erythrocytic development duration via serpentine receptor 10 to coordinate with host rhythms. *Nature communications* **11**(1), 2763. (doi:10.1038/s41467-020-16593-y).
6. Prior K.F., Rijo-Ferreira F., Assis P.A., Hirako I.C., Weaver D.R., Gazzinelli R.T., Reece S.E. 2020 Periodic Parasites and Daily Host Rhythms. *Cell Host Microbe* **27**(2), 176-187. (doi:<https://doi.org/10.1016/j.chom.2020.01.005>).
7. Smith L.M., Motta F.C., Chopra G., Moch J.K., Nerem R.R., Cummins B., Roche K.E., Kelliher C.M., Leman A.R., Harer J. 2020 An intrinsic oscillator drives the blood stage cycle of the malaria parasite Plasmodium falciparum. *Science* **368**(6492), 754-759.
8. Westwood M.L., O'Donnell A.J., de Bekker C., Lively C.M., Zuk M., Reece S.E. 2019 The evolutionary ecology of circadian rhythms in infection. *Nature ecology & evolution*, 1.
9. Garson G.D. 2019 *Multilevel Modeling*, SAGE Publications.
10. Ogwan'g R.A., Mwangi J.K., Githure J., Were J., Roberts C.R., Martin S.K. 1993 Factors affecting exflagellation of in vitro-cultivated Plasmodium falciparum gametocytes. *The American journal of tropical medicine and hygiene* **49**(1), 25-29.
11. O'Donnell A.J., Mideo N., Reece S.E. 2014 Erratum to: disrupting rhythms in Plasmodium chabaudi: costs accrue quickly and independently of how infections are initiated. *Malar J* **13**(1), 503.
12. O'Donnell A.J., Mideo N., Reece S.E. 2013 Disrupting rhythms in Plasmodium chabaudi: costs accrue quickly and independently of how infections are initiated. *Malar J* **12**, 372. (doi:10.1186/1475-2875-12-372).
13. O'Donnell A.J., Schneider P., McWatters H.G., Reece S.E. 2011 Fitness costs of disrupting circadian rhythms in malaria parasites. *Proc Biol Sci* **278**(1717), 2429-2436. (doi:rsob.2010.2457 [pii] 10.1098/rsob.2010.2457).

Appendix E

Comments are below by reviewer 3.

Summary: The main weaknesses of the paper remain experiments that are only 32 hours and having to specify the period a priori and not from the data. Also there are no molecular correlates in the parasite, such as gene expression, to follow rhythmicity in the parasite and rhythmicity in the feeding schedule of the host. The paper has been strengthened by a discussion of mechanisms for the alignment of parasite IDC with host feeding rhythm. The cited Science paper by Takahashi does not find a strong link between host feeding schedule and periodicity in gene expression in the parasite (see Fig. 3 in science). This is a contrast to the results of the authors on IDC. The reader might want to know how Fig. 3D in Takahashi relates to IDC in the authors' work. I am going to recommend publication.

Dear Professor Heesterbeek

We are grateful for the opportunity to revise the manuscript (RSPB-2020-0347.R1) and thank the reviewers for their continued attention to it. Both reviewer 1 and 2 are happy with the manuscript as it stands and we have made extensive changes in response to reviewer 3's comments.

We have never had a paper cause so much confusion with a reviewer before. We think we have finally identified a sentence in the abstract and the use of terms from evolutionary ecology that can be interpreted by readers from other disciplines in ways that are not our intention. We think this explains the disconnect between our intention and Reviewer 3's interpretation. Thus, we have altered the abstract, intro, and discussion to clarify our aims, by using terminology that should be clear across disciplines.

The central issue that the remaining reviewer (#3) has, stems from their belief that we dismiss the notion that malaria parasites might possess an endogenous clock. Identifying clocks and how they function are core activities in chronobiology and so we suspect this may be reviewer 3's perspective. Put another way, it seems the reviewer assumes our paper is about testing for the presence of a parasite clock. However, we have been careful to avoid speculation about the mechanism underpinning rhythmicity in the parasite because our data do not directly speak to this issue. Our experiments are designed to determine which kind of host rhythm the parasite's rhythm is aligned with, not how this alignment is achieved. We think this misunderstanding also explains why the reviewer is critical of our approach – we agree our approach is not the right way to test for a clock, but as that is not our aim, our approach is not flawed. We have again explained this in the response letter and conveyed our aims more clearly in the revised manuscript.

We are aware that another group are preparing a paper with very similar experiments (just using a different kind of clock mutant mouse host for malaria parasites) and they find the same results. Our paper was first submitted to Proc Roy Soc B back in September 2019 and has experienced long delays at each stage whilst out to review. Thus, we would be grateful if the Editor/ Associate Editor are able to make a decision without the time costs involved in soliciting further reviews.

Many thanks,

Aidan O'Donnell, Kim Prior, & Sarah Reece

Reviewer 3:

The reviewer's central concern is that we conclude malaria parasites do not have a circadian clock, and that this conclusion is not clearly supported by our data.

Our experiments are designed to determine which kind of host rhythm the parasite's rhythm is aligned with, not how this alignment is achieved (i.e. whether by a parasite clock or any other scheduling method). This also explains why the reviewer is critical of our approach – we agree our approach is not the right way to test for a clock, but, as we explained in our previous response, that is not our aim. However, we now realise that a sentence in the abstract and some of the terms we used could be interpreted in different ways. For example, when we have used the term "driven" we do not

mean to suggest that the host is imposing a rhythm upon intrinsically arrhythmic parasites by killing out-of-synch IDC stages.

To remove this confusion, we have substantially re-written the abstract, introduction, and discussion to convey our aims and conclusions more clearly, including a change in terminology. This includes specifying throughout whether we are referring to a parasite or host rhythm when use the terms feeding-related rhythms and clock-driven rhythms. The re-write also emphasizes that our approach is designed to probe the nature of the interaction between host and parasite rhythms, not the type of mechanism(s) that underpins this interaction. E.g. we now write *“We test if rhythmicity in parasite replication is coordinated with the host’s feeding-related rhythms and/or rhythms driven by the host’s canonical circadian clock”*. And *“We conclude that malaria parasite replication is coordinated to rhythmic host processes that are independent of the core-clock proteins PERIOD 1 and 2; mostly likely a periodic nutrient made available when the host digests food.”* However, seeing as the reviewer desires discussion of whether the parasites have a clock or not, we have edited the conclusions section to address to what extent our data are consistent with parasites keeping time themselves, via a clock or simpler response to time-of-day. *“The speed with which the IDC schedule changes, its precision, and the modest initial loss of parasite number involved in rescheduling [12] along with parasite control of IDC duration [24], suggest the parasite is actively aligning certain developmental stages with host feeding rhythms to take advantage of periodicity in a resource(s) it must acquire from the host’s processing of its food. Recent studies suggest that IDC rhythms display a hallmark of a circadian clock [9,10] but other criteria (temperature compensation, entrainment) are yet to be met. Whatever the parasites’ method of time-keeping, our data suggest it uses a signal stemming from the host’s processing of food as a Zeitgeber or timing cue.”*

Response: I would state things slightly differently. This reviewer thinks that some consideration should be given to the hypothesis of a clock in Plasmodium to explain the data. The paper is stronger by at least connecting to possible mechanisms including a circadian rhythm in the parasite. The paper is also strengthened by virtue of citing the Science papers.

Responses to other comments:

- 1) *“The standard size of an experiment is 48 h. Examples include any of the papers by Jay Dunlap and colleagues. See, for example, Crosthwaite et al. (1995) in Cell 81: 1004 or in Science 276: 763 (1997). The problem with the shorter period for the experiment is the inability to examine period variation. For example, in the JTK_CYCLE and harmonic regression analysis the authors had to fix the period at 24 h. The period is relevant in distinguishing a feeding rhythm from a circadian rhythm in the parasite to explain the observations on IDC. Also the phase is the noisiest part of the characterization usually and is part of their paper’s point about synchronization. Having more time points will strengthen the synchronization argument. Harmonic regression is actually meant for much longer time series. See, for example, F. J. Anscombe’s Computing in Statistical Science with APL. In this work in Ch. 10 harmonic regression is used for 100s if not 1000s of observations. This is the appropriate context for testing the normality assumption underlying harmonic regression using Wilks-Shapiro*

The issue here is that *“The period is relevant in distinguishing a feeding rhythm from a circadian rhythm in the parasite”*. As explained above, we are not attempting to distinguish between the parasite using a ‘circadian clock (or other kind of time-keeping mechanism) to couple its IDC to the timing of host feeding’ versus ‘a nutrient from the host’s food enforcing a rhythm upon the IDC via mistimed parasites starving to death’. Our data, across our treatment groups and in both experiments, reveal that the IDC rhythm is aligned – by an unknown mechanism – to a rhythm stemming from the timing of host feeding alone, and not to a host rhythm that depends on the host’s canonical circadian clock. Put another way, if the parasite has a clock (or other time-keeping mechanism) then it is using a signal provided by the host’s processing of food as a Zeitgeber (time-cue). The conclusions section now states: *“Whatever the parasites’ method of time-keeping, our data suggest it uses a signal stemming from the host’s processing of food as a Zeitgeber or timing cue”*. In contrast, if we were probing for the presence of a clock then yes, we would examine a much longer time-series because estimating period would be necessary (which would tradeoff against the frequency of samples that could be taken). We also note that other studies provide ample evidence that the period of the IDC matches that of the host’s rhythm [1-5]. Furthermore, revealing that the period of the IDC matches that of host rhythms (i.e. is approx. 24h) would not help differentiate which host rhythm the IDC is aligned with because the feeding rhythms we impose have a period of 24 hours and WT hosts have circadian rhythms with a period of 24 hours. We designed our study to decouple the

timing (phase) of feeding and TTFL-clock rhythms in the host, and thus, use phase and synchrony of the IDC to test which host rhythms it aligns with.

In response to “*Also the phase is the noisiest part of the characterization usually and is part of their paper’s point about synchronization*”, we urge the reviewer to consult the results section again. We do not use phase as a measure of synchrony. We report phase (timing of the parasites rhythm) in relation to the phase of the host’s feeding rhythms and we report amplitude as a measure of synchrony of the parasite rhythm. Moreover, that the phase of the parasite rhythm aligns with host feeding rhythms across our treatment groups and in both experiments, despite its “noise” makes our results conservative.

Response: Not knowing the period makes the results weaker because it prevents the authors from saying something about mechanism. It also weakens the degree of support for the findings of the paper. If one achieves the same conclusion without making an assumption about period, that is a stronger finding. Not using a standard duration of the experiment of 48 h also makes the paper weaker. What saves the authors is the reported R².

- 2) **The authors put forward the hypothesis that rhythmic feeding is driving the oscillations in the parasites. An alternative equally plausible hypothesis is that the parasite has circadian rhythms to anticipate rhythmic availability of food. That is why it is relevant to know something about molecular correlates in the parasite.**

As explained above and in the general comments, our aim is not to speculate on **how** the parasite IDC is aligned with host feeding rhythms (i.e. the underlying parasite mechanisms). It seems that the reviewer has interpreted our conclusion to be that the parasite is intrinsically arrhythmic and that the host is enforcing a rhythm on the IDC, by for example, starvation of mistimed IDC stages. This might be the reality – although we doubt it is because most evidence, including from our lab [4-8] suggests the parasite is at least in part responsible for scheduling its own IDC. Instead, we propose that if the parasite has a clock (or other time-keeping mechanism) then it is using a signal provided by the host’s processing of food as a Zeitgeber (time-cue). We have altered the conclusion section to make it clear that the parasite probably has some way to tell the time but this cannot be assumed based on our data or that of recent papers. “*The speed with which the IDC schedule changes, its precision, and the modest initial loss of parasite numbers involved in rescheduling [12] along with parasite control of IDC duration [24], suggest the parasite is actively aligning certain developmental stages with host feeding rhythms to take advantage of periodicity in a resource(s) it must acquire from the host’s processing of its food. Recent studies suggest that IDC rhythms display a hallmark of a circadian clock [9,10] but other criteria (temperature compensation, entrainment) are yet to be met.*”

More generally, it is not clear what the reviewer means by “molecular correlates” (“*That is why it is relevant to know something about molecular correlates in the parasite.*”). Demonstrating rhythmicity in expression, or at the protein level, of clock genes would – of course - indicate a clock, but this is circular because it first requires identifying clock genes. Alternatively, if the reviewer means the common approach used in chronobiology to generate a reporter cell line (e.g. using a luciferase tag) as a readout of a clock, then this would be a readout of a potential clock, but likewise, the IDC phenotype is also an output of a potential clock. Given that our aim is to explain the IDC phenotype, it is not clear what would be gained from studying this trait indirectly. Furthermore, recent papers [4, 5, 7] provide plenty of evidence that gene expression of the parasites are rhythmic (with a period that matches the host’s period) – but no clock genes have been identified from these hits and the overwhelming majority are rhythmic simply due to the temporal cascade of developmental changes during the IDC.

Response: What would be gained is understanding the mechanism.

- 3) **The reason that this question is raised is simply to know how general the findings are in the paper.**

We believe this comment refers to the first review in which the reviewer queried whether our findings are generalizable across strains. We addressed this in the revision and added the sentence “*We expect our findings to generalise across strains given the similarities in the IDC schedules observed in this study and in Hirako et al 2018 [18] and Prior 2018 [31] which used different strains of P. chabaudi.*” to the paper. It is not clear to us whether the reviewer is justifying why they queried this in the first review or if they are not satisfied by our addition to the paper.

- 4) The problem is that a circadian oscillator in the parasite could explain the authors' data as an alternative hypothesis; therefore, the period observed is highly relevant to distinguishing a feeding rhythm hypothesis versus a circadian rhythm in the parasite hypothesis.

Please refer to the general comment and responses to points (1) and (2).

- 5) The *P. knowlesi* case is what reviewer 3 had in mind. The reviewer wanted to be sure that the author's choice of Plasmodium in this host did not fall in the same category as *P. knowlesi*. That the authors' choice of strain/host combination does not fall in this category should probably be mentioned in the text.

We have now included reference to 3 recent papers [4, 5, 7] that reveal the same principles underpin rhythmicity of the IDC in human and rodent parasites, including the species used in our paper.

- 6) How can the authors talk about synchronized feeding driving the oscillations seen in the parasite without entertaining the alternative hypothesis that the parasite has a circadian rhythm anticipating such feeding schedules? It is in this context that having more time points is relevant. Being able to estimate period of IDC then becomes relevant for distinguishing these two hypotheses.

Please refer to the general comment and responses to points (1) and (2).

Specific Comments (L is line number)

L25. The authors cannot rule out the role of a possible parasite circadian rhythm. L38. It is interesting that the authors discuss circadian rhythms in the pathogen *Botrytis cinerea*. Why doesn't this example support examining the same hypothesis here? L69. How do the parasites schedule their IDC? It could be using a parasite clock or an environmental cue, such as feeding. L90. It is hard to test the role of environmental cues without entertaining the effects of circadian rhythm anticipating these cues.

We have combined these comments as they all stem from the same idea that parasites may have an intrinsic oscillator. We agree that this is possible but as explained previously and above, our intention is not to uncover the mechanism that is driving the parasite rhythm. Rather our paper reveals that host feeding is a cue for the IDC schedule, either in the form of a Zeitgeber for an intrinsic parasite clock or as a time-of-day cue for a simpler "just-in-time" reactionary mechanism.

Response: Entertaining both hypotheses makes the paper stronger in the conclusion.

L198. The chi-squared statistic makes no sense here with 15 df. It should not have a $P < .001$. The mean of the chi-squared statistic under the null is around 15. So, a chi-squared of .43 is well below the cutoff of 25.00 at the 0.05 level. L211. I have similar problems with the chi-squared statistics reported here. They are very large in magnitude with respect to the df and should be significant at any reasonable alpha level. I have similar problems in the supplements. L217 Same problem here with too large a chi-squared statistic reported to be considered insignificant. L222 The chi-squared statistic is small, but it is reported as significant. Don't understand what is going on here.

Previously, we reported the change in deviance (the difference in $2 \times (\log \text{likelihood})$ between a saturated model and the actual model) as these values are synonymous with a chi-square value when comparing generalised linear models [9]. We apologise that we were not explicit in pointing out that we were reporting deviance, and we agree that this can make it difficult to interpret the goodness of fits for the models (as the deviance scales with the data). We have re-analysed the models using F-tests and have made interpretation of goodness of fits more intuitive by reporting F-values throughout the manuscript (with the exception of the ESM in which Chi-square values for linear mixed effect models with mouse ID as a random effect were now used).

L257. It would be nice to know that the results do generalize across strains.

Please refer to the response to point (3).

L281. Normally in mammals there is glucose compensation, although one might argue that this phenomenon

may not make sense in a microbe. I think the authors make the key point that even if there is glucose compensation normally, the presence of the parasite may shut down this mechanism. So, glucose signaling could be involved as an environmental signal.

The idea that glucose could provide environmental time of day information to the parasite is proposed by Prior et al [3] and Hirako et al [1]. These studies investigated well-developed infections and the glucose rhythm observed relies on highly active immune responses. Thus, a strong glucose rhythm is unlikely to operate at the start of infections, before immune responses are well developed. Given that the IDC can become rescheduled in the first few days of infections, it is unlikely that glucose is responsible – either by enforcing an IDC rhythm and/or by providing time-of-day information to the parasite [6].

L320. It would be interesting to test whether or not temperature could be used as an entrainment signal.

We provided an extensive rebuttal of this possibility in the first revision and included new figures and a dedicated section in the discussion section of the manuscript. Our reasoning for ruling out temperature is copied from the previous rebuttal below, but it is not clear whether the reviewer disagrees (and if so, with which aspects).

“We have dedicated a paragraph to discussion of body temperature as a Zeitgeber/time-of-day cue (lines 301-316).

“We agree that we do see temperature rhythms in our TRF mice that is close to a full inversion when compared to WT mice (ESM Figure S3b). This differs from Prior et al [3] in which no complete inversion was seen, perhaps due the differences in methodology e.g. mice strain (Prior used MF1 vs our C57/bl6), housing (Prior group housed animals while we housed mice individually) and differences in TRF timing (Prior provided food for 12 hours vs our 10 hours). However, it is well known that malaria parasites have an ability to respond to temperature changes (gametogenesis in transmission stages is partly triggered by temperature) [10]. Yet, despite these points, we doubt host temperature drives parasite rhythms because responding to absolute temperature in the mammalian host would only give parasites a 12hour resolution to their timekeeping (i.e. cold in the day vs warm at night), and thus we would expect to see lower stage amplitudes. Alternatively, parasites may respond to a sharp change in temperature (either using an increase or decrease in host temperature as a timing cue). This creates two scenarios, both of which we suspect are unlikely:

1) *Parasites are scheduled by a sharp increase in temperature: This is unlikely to be a reliable signal to parasites as there are several occasions in the 24hr period when temperature increases occur. For example, in ad lib fed WT mice, which are more analogous to natural conditions than the other groups, (ESM Figure S3b) temperature rises sharply in the early part of the light period and again early in the dark period. Thus, if parasites were scheduled solely by an increase in temperature, we would expect infections to stochastically follow the light or dark increases in temperature, resulting in 50% of infections in this treatment group having each, opposing, schedule. Instead we observe very repeatable parasite rhythms within in this group.*

2) *Parasites are scheduled by a sharp decrease in temperature: Again, this is unlikely to be a reliable signal. In ad lib WT mice, there are two drops in temperature spaced 3 hours apart during the light period, and in TRF WT mice there are two significant drops in temperature spaced 12 hours apart (early in the light period and early in the dark period; ESM Figure S3b).*

It is theoretically possible that parasites “ignore” a cue when they are at certain stages in the IDC, which enables them to respond to the “correct” cue at the right time, but this is not consistent with observations. For example, this strategy cannot explain the rescheduling of parasites that have been mismatched by 12 hours to the host’s rhythm, because the parasites wouldn’t “know” that the cue appearing in time with their IDC stage is incorrect and thus never reschedule ([3, 11-13] and this study in which parasites in TRF mice were initiated out of phase with the feeding rhythm).

Given that the relevance of temperature rhythms to the IDC schedule is remote, we do not feel it appropriate to spend more than a paragraph discussing it and body temperature rhythms are presented in detail in Figures S1, S2 and S3b.

References:

1. Hirako I.C., Assis P.A., Hojo-Souza N.S., Reed G., Nakaya H., Golenbock D.T., Coimbra R.S., Gazzinelli R.T. 2018 Daily Rhythms of TNF α Expression and Food Intake Regulate Synchrony of Plasmodium Stages with the Host Circadian Cycle. *Cell Host Microbe* **23**(6), 796-808 e796. (doi:10.1016/j.chom.2018.04.016).
2. Prior K.F., Rijo-Ferreira F., Assis P.A., Hirako I.C., Weaver D.R., Gazzinelli R.T., Reece S.E. 2020 Periodic Parasites and Daily Host Rhythms. *Cell Host Microbe* **27**(2), 176-187. (doi:10.1016/j.chom.2020.01.005).
3. Prior K.F., van der Veen D.R., O'Donnell A.J., Cumnock K., Schneider D., Pain A., Subudhi A., Ramaprasad A., Rund S.S.C., Savill N.J., et al. 2018 Timing of host feeding drives rhythms in parasite replication. *PLoS Pathog* **14**(2).
4. Rijo-Ferreira F., Acosta-Rodriguez V.A., Abel J.H., Kornblum I., Bento I., Kilaru G., Klerman E.B., Mota M.M., Takahashi J.S. 2020 The malaria parasite has an intrinsic clock. *Science* **368**(6492), 746-753.
5. Subudhi A.K., O'Donnell A.J., Ramaprasad A., Abkhallo H.M., Kaushik A., Ansari H.R., Abdel-Haleem A.M., Ben Rached F., Kaneko O., Culleton R., et al. 2020 Malaria parasites regulate intra-erythrocytic development duration via serpentine receptor 10 to coordinate with host rhythms. *Nature communications* **11**(1), 2763. (doi:10.1038/s41467-020-16593-y).
6. Prior K.F., Rijo-Ferreira F., Assis P.A., Hirako I.C., Weaver D.R., Gazzinelli R.T., Reece S.E. 2020 Periodic Parasites and Daily Host Rhythms. *Cell Host Microbe* **27**(2), 176-187. (doi:<https://doi.org/10.1016/j.chom.2020.01.005>).
7. Smith L.M., Motta F.C., Chopra G., Moch J.K., Nerem R.R., Cummins B., Roche K.E., Kelliher C.M., Lemansky A.R., Harer J. 2020 An intrinsic oscillator drives the blood stage cycle of the malaria parasite Plasmodium falciparum. *Science* **368**(6492), 754-759.
8. Westwood M.L., O'Donnell A.J., de Bekker C., Lively C.M., Zuk M., Reece S.E. 2019 The evolutionary ecology of circadian rhythms in infection. *Nature ecology & evolution*, 1.
9. Garson G.D. 2019 *Multilevel Modeling*, SAGE Publications.
10. Ogwan'g R.A., Mwangi J.K., Githure J., Were J., Roberts C.R., Martin S.K. 1993 Factors affecting exflagellation of in vitro-cultivated Plasmodium falciparum gametocytes. *The American journal of tropical medicine and hygiene* **49**(1), 25-29.
11. O'Donnell A.J., Mideo N., Reece S.E. 2014 Erratum to: disrupting rhythms in Plasmodium chabaudi: costs accrue quickly and independently of how infections are initiated. *Malar J* **13**(1), 503.
12. O'Donnell A.J., Mideo N., Reece S.E. 2013 Disrupting rhythms in Plasmodium chabaudi: costs accrue quickly and independently of how infections are initiated. *Malar J* **12**, 372. (doi:10.1186/1475-2875-12-372).
13. O'Donnell A.J., Schneider P., McWatters H.G., Reece S.E. 2011 Fitness costs of disrupting circadian rhythms in malaria parasites. *Proc Biol Sci* **278**(1717), 2429-2436. (doi:rsob.2010.2457 [pii] 10.1098/rsob.2010.2457).

Appendix F

Comments are below by reviewer 3.

Summary: The main weaknesses of the paper remain experiments that are only 32 hours and having to specify the period a priori and not from the data. Also there are no molecular correlates in the parasite, such as gene expression, to follow rhythmicity in the parasite and rhythmicity in the feeding schedule of the host. The paper has been strengthened by a discussion of mechanisms for the alignment of parasite IDC with host feeding rhythm. The cited Science paper by Takahashi does not find a strong link between host feeding schedule and periodicity in gene expression in the parasite (see Fig. 3 in science). This is a contrast to the results of the authors on IDC. The reader might want to know how Fig. 3D in Takahashi relates to IDC in the authors' work. I am going to recommend publication.

Dear Professor Heesterbeek

We are grateful that our manuscript (RSPB-2020-0347.R2) has been accepted for publication and thank the reviewers for their continued attention to it. In particular, we are happy that Reviewer 3 feels the paper is stronger after our edits and has recommended publication.

We have made minor changes to the discussion (as recommended by Reviewer 3) which now includes a paragraph detailing how our results appear to differ from those published recently in Science by Rijo-Ferreira (2020; Takahashi lab). Specifically, the experiment mentioned by the reviewer tests whether the act of eating, not the downstream consequences of food digestion, schedules the parasite IDC. This distinction is subtle but important, because overall, the results presented in this paper are in keeping with ours.

Many thanks,

Aidan O'Donnell, Kim Prior, & Sarah Reece